# Advancing Counterfactual Prediction through Nonlinear Quantile Regression

**Shaoan Xie**\*                                                                                    *shaoan@cmu.edu*
*Carnegie Mellon University*
*Mohamed bin Zayed University of Artificial Intelligence*

**Biwei Huang**\*                                                                                    *bih007@ucsd.edu*
*University of California San Diego*

**Bin Gu**                                                                                            *gubin@jlu.edu.cn*
*Jilin University*

**Tongliang Liu**                                                                        *tongliang.liu@sydney.edu.au*
*The University of Sydney*

**Peter Spirtes**                                                                              *ps7z@andrew.cmu.edu*
*Carnegie Mellon University*

**Kun Zhang**                                                                                        *kunz1@cmu.edu*
*Carnegie Mellon University*
*Mohamed bin Zayed University of Artificial Intelligence*

**Reviewed on OpenReview:** *https://openreview.net/forum?id=0T1Pd65fyD*

## Abstract

The ability to answer counterfactual "what if" questions is essential for understanding and leveraging causal relationships. Traditional counterfactual prediction under Pearl's framework typically relies on access to, or estimation of, a structural causal model (SCM), which is often unavailable and difficult to identify in practice. While prior work has explored rank- and quantile-preserving approaches as an alternative, existing methods often rely on instrumental variables, high-dimensional density estimation, or optimal transport constructions, limiting their applicability in practice. In this paper, we develop a neural bi-level optimization framework that directly operationalizes the quantile-preservation principle for counterfactual prediction. The proposed bi-level framework learns the latent quantile associated with an observation and predicts counterfactual outcomes by preserving this quantile under intervention. We provide a theoretical analysis of the resulting formulation, establishing uniqueness of the population-level solution under mild assumptions and deriving finite-sample generalization guarantees for the conditional quantile estimator. Empirical evaluations across multiple datasets demonstrate the effectiveness of the proposed approach and support the theoretical results.

## 1 Introduction

Understanding and making use of cause-and-effect relationships play a central role in scientific research, policy analysis, and everyday decision-making. Pearl's causal ladder (Pearl, 2000) delineates the hierarchy of prediction, intervention, and counterfactuals, reflecting their increasing complexity and difficulty. Counterfactual prediction, the most challenging level, allows us to explore what would have happened if certain

---

\*Equal contribution.

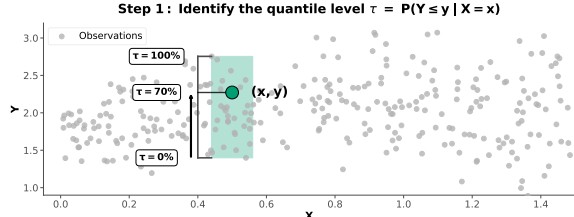 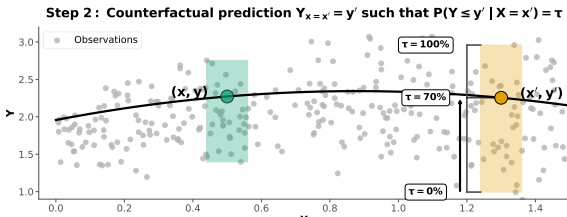

Figure 1: **Illustration of counterfactual prediction through quantile preservation.** Given a structural causal model $Y = f_Y(X, Z, r(E))$, where both $f_Y$ and $r(E)$ are unknown, if $Y$ is strictly monotonic in $r(E)$, then for a factual observation $y$ and its counterfactual outcome $Y_{X=x'}$, we have $P(Y \leq y \mid X = x, Z = z) = P(Y \leq Y_{X=x'} \mid X = x', Z = z)$ under the same noise term $r(e)$. Thus, for a factual sample $(X = x, Z = z, Y = y)$, we first estimate the quantile $\tau = P(Y \leq y \mid X = x, Z = z)$ (e.g., $\tau = 0.70$), and then obtain the counterfactual value $Y_{X=x'} = y'$ such that $P(Y \leq y' \mid X = x', Z = z) = \tau$. For clarity, $Z$ is omitted from the illustration.

actions or conditions had been set to a different value, providing valuable insights into the underlying causal relationships between variables.

Conventional approaches to counterfactual prediction often rely on having access to or estimating a structural causal model (SCM). For instance, within Pearl's framework (Pearl, 2000), if the SCM is given, the typical three-step procedure involves first estimating the noise value of the sample of interest, then modifying the model, and finally computing the counterfactual value using the estimated noise within the adjusted SCM. Unfortunately, the SCM is often unavailable and general SCMs may be non-identifiable (Zhang & Hyvarinen, 2012; Zhang et al., 2015). Consequently, to estimate the full SCM, many approaches either impose assumptions on the noise distribution and the functional form of the target-generating process to ensure identification, or operate without identification guarantees. For instance, Khemakhem et al. (2021) assume affine generating functions with Gaussian noise and provides identification guarantees. By contrast, Sanchez & Tsaftaris (2022); Pawlowski et al. (2020) avoid strong assumptions on the data-generating function; they first train an invertible generative model for the target variable and then invert it to recover the noise values. However, these methods lack theoretical guarantees for the resulting counterfactual outcomes.

These challenges have motivated alternative approaches that avoid explicit recovery of the full SCM. In particular, a growing line of work has shown that counterfactual outcomes can be characterized through the preservation of latent ranks or conditional quantiles under suitable assumptions (Chernozhukov & Hansen, 2005; Chernozhukov et al., 2017; Charpentier et al., 2023; Machado et al., 2025; Plečko & Meinshausen, 2020; De Lara et al., 2021; Wu et al., 2025). Under this perspective, when the outcome is monotonic with respect to the unobserved noise, each noise realization determines the outcome's relative position within the corresponding conditional distribution, which can be represented by a conditional quantile level. Because the latent noise remains invariant under intervention, this quantile level is preserved. Consequently, the counterfactual outcome can be obtained by evaluating the same quantile level under the intervened distribution without explicitly recovering the noise term or specifying the functional form of the data-generating process.

Building on this characterization, prior work has pursued several approaches to identifying or estimating counterfactual outcomes. Chernozhukov & Hansen (2005); Chernozhukov et al. (2017) establish rank invariance for identifying quantile treatment effects using instrumental variables and structural equations; however, the resulting inverse-inference procedure typically requires a search over the parameter space and depends on the availability of valid instrumental variables. Plečko & Meinshausen (2020) formulate counterfactual prediction through quantile preservation, but assume access to the underlying data-generating densities. Charpentier et al. (2023); De Lara et al. (2021); Machado et al. (2025) employ optimal transport and distributional coupling to construct counterfactual mappings; however, such formulations can be difficult to implement and interpret in practice (Machado et al., 2025). More recently, Wu et al. (2025) estimate counterfactual outcomes through conditional density estimation, but its performance depends on accurate high-dimensional density estimation, which can be challenging in practice (see Table 1).

Motivated by these limitations, we develop a neural bi-level optimization framework that directly operationalizes the quantile-preservation principle for counterfactual prediction. As illustrated in Fig. 1, the proposed framework estimates the latent quantile associated with a factual observation and predicts the counterfactual outcome by evaluating the corresponding quantile under intervention. To achieve this, we reformulate counterfactual prediction as a bi-level quantile regression problem. The lower level learns conditional quantile functions that characterize the population-level outcome distribution, while the upper level identifies the latent quantile associated with an observed factual outcome. Importantly, we provide a theoretical analysis of the proposed formulation, showing that the population-level bi-level optimization problem admits a unique solution under the stated assumptions and deriving finite-sample generalization guarantees for the learned conditional quantile estimator.

**Our main contributions are summarized as follows:**

- We propose a **neural bi-level quantile regression framework** for counterfactual prediction. By directly operationalizing the quantile-preservation principle through a bi-level optimization objective, the proposed framework estimates counterfactual outcomes without requiring instrumental variables, density estimation, or explicit optimal transport constructions.

- We provide a **theoretical analysis** of the proposed formulation. We establish **uniqueness** of the population-level solution under the stated assumptions and derive **finite-sample generalization guarantees** for the learned conditional quantile estimator.

- We conduct **extensive empirical evaluations** across diverse tabular and image-based counterfactual prediction tasks. The results demonstrate that the proposed framework consistently achieves competitive or superior performance compared with existing approaches while exhibiting strong sample efficiency and robustness across a range of settings.

## 2 Problem Formulation and Related Work

In this section, we introduce key concepts relevant to our study, including Pearl's three-step procedure for counterfactual prediction, the technique of quantile regression, and recent works in counterfactual prediction under Pearl's procedure. Below, we first give a formal definition of counterfactual outcomes.

**Definition 1** (Counterfactual outcome )**.** Following the structural causal model framework of Pearl (2000), suppose $X$, $Y$, and $Z$ are random variables, where $X$ and $Z$ are direct causes of $Y$. Given an observed factual instance $\langle X = x, Y = y, Z = z \rangle$, the counterfactual outcome is the random variable $Y_{x'}$, which denotes the value that $Y$ would have taken had $X$ been set to a different value $x'$ while all other factors remained unchanged.

**Pearl's Three-Step Procedure for Counterfactual Prediction** In the context of SCMs, Pearl (2000); Pearl et al. (2016) introduced a three-step procedure—abduction, action, and prediction—to reason about counterfactuals.

Suppose an SCM $M$ is given by the structural equations:

$$Y = f_Y(X, Z, E_Y), \quad X = f_X(Z, E_X), \quad Z = f_Z(E_Z),$$

where $E = (E_Y, E_X, E_Z)$ represents the exogenous (noise) variables. Given an observed factual instance $\langle X = x, Y = y, Z = z \rangle$, counterfactual inference of the outcome $Y$ under the hypothetical intervention $X = x'$ proceeds as follows:

- **Step 1 (Abduction):** Use the observed evidence $\langle X = x, Y = y, Z = z \rangle$ to infer the posterior distribution of the exogenous variables $E$ (specifically $P(E|x, y, z)$).

- **Step 2 (Action):** Modify the model $M$ by replacing the structural equation for $X$ with the intervention constant $X = x'$, yielding the submodel $M_{x'}$.

- **Step 3 (Prediction):** Use the modified model $M_{x'}$ and the posterior distribution of $E$ obtained in the abduction step to compute the counterfactual outcome $Y_{x'}$.

**Counterfactual Prediction** Various deep-learning approaches have been proposed for estimating the SCM and noise values using observational data and accordingly perform the three-step procedure for counterfactual prediction. Khemakhem et al. (2021) propose to use autoregressive flow to perform causal discovery by comparing the likelihood and infer the noise for counterfactual prediction by inverting the flow. Javaloy et al. (2023) present a mechanism to embed additional causal knowledge when learning the causal autoregressive flow. CFQP (De Brouwer, 2022) considers a setting when the background variables are categorical and employs the Expectation-Maximization framework to predict the cluster of the sample and perform counterfactual prediction with the regression model trained on the specific cluster. CTRL (Lu et al., 2020a) and BGM (Nasr-Esfahany et al., 2023) show that the counterfactual outcome is identifiable when the SCM is monotonic w.r.t. the noise term. In particular, BGM uses conditional spline flow to mimic the generation process and performs counterfactual prediction by reversing the flow. DeepSCM (Pawlowski et al., 2020) proposes to use variational inference and normalizing flow to infer the noise variable. DiffSCM (Sanchez & Tsaftaris, 2022) proposes to match the observation distribution with a conditional diffusion model and infer the noise by reversing the diffusion process, but it only allows one cause. DCM (Chao et al., 2023) generalizes DiffSCM to support multiple causes of a single variable. G-NCM (Xia et al., 2022) extends the neural causal model (Xia et al., 2021) to estimate the counterfactual distribution. Melnychuk et al. (2022) propose a domain confusion loss to address confounding bias and use a transformer to perform counterfactual prediction for long-range time-series data. Tsirtsis & Rodriguez (2024) develop a search method to find the counterfactually optimal sequences when the state space is continuous. Ribeiro et al. (2023) propose a hierarchical latent mediator model for counterfactual image generation task.

A related line of work approaches counterfactual prediction through rank or quantile preservation rather than explicit recovery of the underlying SCM. Several works share the core principle of preserving quantiles or utilizing transport mappings for counterfactual prediction. Chernozhukov & Hansen (2005); Chernozhukov et al. (2017) establish rank invariance to identify quantile treatment effects using instrumental variables and explicit structural equations. However, their methodology relies on an inverse-inference procedure that typically requires a grid search over the parameter space, which is computationally prohibitive in high-dimensional settings. In contrast, we propose a direct neural bi-level optimization framework. Our approach learns quantile levels directly from observational data without specifying a structural model, yielding precise, point-valued individual counterfactual predictions via scalar quantile preservation. Plečko & Meinshausen (2020) propose quantile preservation but assume access to the underlying data-generating densities. Alternatively, Charpentier et al. (2023), De Lara et al. (2021), and Machado et al. (2025) rely on optimal transport theory—using it to justify quantile matching, formulate probabilistic distributional couplings, or construct sequential causal mappings, respectively. Instead of relying on assumed densities or explicit optimal transport constructions, we propose a direct neural bi-level estimation framework. Our approach learns quantile levels directly from data without specifying the model, yielding precise, point-valued counterfactual predictions via scalar quantile preservation. Recent work RankPrev (Wu et al., 2025) performs counterfactual prediction by preserving the rank information, however, it only supports binary interventions and has to rely on a kernel density estimator which can be inaccurate with finite training data.

Individual treatment effect (ITE) research (Johansson et al., 2016; Yoon et al., 2018; Bica et al., 2020; Yao et al., 2018; Li & Yao, 2022; Lu et al., 2020b; Zhou et al., 2021b;a) is also deeply connected to counterfactual prediction while the former focuses on the differences between expected outcomes over the population before and after intervention. We present more ITE works in the Appendix A.

**Quantile Regression** Traditional regression estimation focuses on estimating the conditional mean of $Y$ given $X$. On the other hand, quantile regression (Koenker & Hallock, 2001) is concerned with estimating conditional quantiles, specifically the $\tau$-th quantile $\mu_\tau$, which is the minimum value $\mu$ such that $P(Y \leq \mu|X) = \tau$, where $\tau$ is a predefined value. Some quantile regression settings encounter the quantile crossing problem where estimated conditional quantile functions for different quantile levels intersect, violating the natural monotonicity of quantiles with respect to their levels. Takeuchi et al. (2009); Tagasovska & Lopez-Paz (2019) propose loss functions to learn all the conditional quantiles of a given target variable with neural networks to address this issue. Fortunately, as our framework only requires learning a single quantile for the observation which contains essential information about the noise term, this issue can be avoided.

## 3   Quantile-Regression-based Counterfactual Prediction: Theoretical Insights

Conventional counterfactual prediction methods typically follow Pearl's three-step procedure, which requires estimating both the SCM and the associated noise variables. However, this process faces two major challenges: general SCMs are often non-identifiable (Zhang et al., 2015), and jointly estimating the SCM and noise is inherently difficult in practice.

In this paper, we show that counterfactual prediction can be formulated as a bi-level quantile regression problem, which allows us to bypass the need for explicitly estimating SCMs and noise terms. Specifically, given observed evidence $\langle X = x, Y = y, Z = z \rangle$, we show that the counterfactual outcome $Y_{x'}$ corresponds to the $\tau$-th quantile of the conditional distribution $P(Y \mid X = x', Z = z)$, where the factual outcome $Y = y$ lies at the $\tau$-th quantile of $P(Y \mid X = x, Z = z)$. Leveraging this insight, counterfactual outcomes can be directly estimated through quantile regression. This fundamental relationship is formalized in the lemma below.

**Lemma 1.** *Suppose $Y$ is generated according to the structural causal model*

$$Y = f(X, Z, r(E)),$$

*where $X$ and $Z$ are direct causes of $Y$, $Z$ is a direct cause of $X$, and $E$ is an exogenous noise variable satisfying $E \perp\!\!\!\perp (X, Z)$. The function $r(\cdot)$ is an arbitrary transformation of $E$, and the structural function $f$ is unknown but assumed to be smooth and strictly monotonic in $r(E)$ for fixed $(X, Z)$.*

*Given an observed factual instance $\langle X = x, Y = y, Z = z \rangle$, define*

$$\tau := P(Y \leq y \mid X = x, Z = z).$$

*Then, under the intervention $do(X = x')$, the counterfactual outcome $Y_{x'}$ corresponds to the $\tau$-th quantile of the conditional distribution $P(Y \mid X = x', Z = z)$.*

The proof is provided in Appendix C. This lemma establishes the identifiability of counterfactual outcomes when the structural function $f$ is strictly monotonic with respect to $r(E)$, which covers a wide range of function classes. Below, we present a compilation of commonly encountered special cases where this condition remains valid.

- Linear causal models: $Y = aX + bZ + r(E)$.

- Nonlinear causal models with additive noise: $Y = f(X, Z) + r(E)$.

- Nonlinear causal models with multiplicative noise: $Y = f(X, Z) \cdot r(E)$.

- Post-nonlinear causal models: $Y = h(f(X, Z) + r(E))$.

- Heteroscedastic noise models: $Y = f(X, Z) + h(X, Z) \cdot r(E)$.

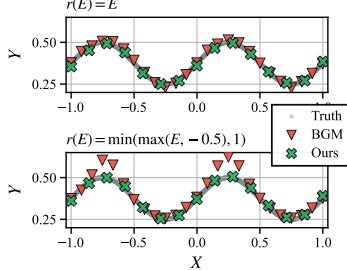

Figure 2: Comparisons under different forms of $r(E)$.

To provide intuition for the monotonicity assumption, consider a clinical setting comparing two weight-loss drugs. If both drugs act through a similar mechanism (e.g., appetite suppression), then patients who lose relatively less weight under drug A will also tend to lose relatively less weight under drug B. In this case, each patient's position in the outcome distribution (their quantile level) is preserved across treatments, which is consistent with our assumption.

In contrast, if the two drugs operate through different mechanisms (e.g., one suppresses appetite while the other increases metabolic rate), patients may respond differently across treatments. A patient who responds poorly to one drug could respond well to the other, leading to changes in their relative position in the outcome distribution. In this case, the quantile level is not preserved, and the assumption underlying our method is violated.

Similar identifiability results under monotonicity assumptions have been studied in prior work (Lu et al., 2020a; Nasr-Esfahany et al., 2023; Tsirtsis & Rodriguez, 2024; Plečko & Meinshausen, 2020; Chernozhukov & Hansen, 2005; Chernozhukov et al., 2017; Charpentier et al., 2023; De Lara et al., 2021; Machado et al., 2025). However, there are two key distinctions that set our approach apart.

**We bypass the need to recover the true noise $E$ and SCM.** Previous works demonstrate identifiability under strict monotonicity of $f$ with respect to the raw noise term $E$ (Lu et al., 2020a; Nasr-Esfahany et al., 2023; Tsirtsis & Rodriguez, 2024). In contrast, we require monotonicity only with respect to a transformed noise representation $r(E)$, which is a strictly weaker and more flexible condition. For example, consider $Y = \frac{1}{4}\big(Z + \sin(2\pi X)\,r(E) + 2r(E)\big)$, with $r_1(E) = E$ and $r_2(E) = \min(\max(E, -0.5), 1)$. In both cases, the outcome is monotonic in $r(E)$ but not necessarily in $E$. Empirically, as shown in Fig.2, baseline method BGM (Nasr-Esfahany et al., 2023), which depends on monotonicity in $E$, exhibits large performance degradation under the second transformation $r_2(E)$, whereas our method remains stable and accurate across both scenarios.

**Our result provides a direct characterization of counterfactual outcomes in terms of conditional quantiles.** Some optimal transport approaches also implicitly preserve quantile or rank information (Charpentier et al., 2023; De Lara et al., 2024), but they typically rely on distributional modeling or transport constructions that can be challenging to implement and interpret in complex settings (Machado et al., 2025). Other methods also characterize counterfactual outcomes with quantiles, but often require access to the underlying data-generating densities (Plečko & Meinshausen, 2020) or rely on instrumental variables (Chernozhukov & Hansen, 2005; Chernozhukov et al., 2017). More closely related works Lu et al. (2020a); Nasr-Esfahany et al. (2023); Tsirtsis & Rodriguez (2024) estimate the full SCM by enforcing monotonicity with respect to the noise variable. However, the noise is often difficult to estimate in practice, and such monotonic constraints can limit the flexibility of neural models. A recent method, RankPrev Wu et al. (2025), instead preserves Kendall's rank correlation, but it assumes binary inputs and relies on kernel smoothing for density estimation, which limits its applicability in continuous settings. Building on the implication of Lemma 1, our approach enables neural networks to jointly learn the latent quantile level and the corresponding conditional quantile function directly from data, without requiring explicit recovery of the noise variable, density estimation, or transport construction. This leads to a flexible and scalable framework, particularly suitable for high-dimensional settings, as demonstrated in Table 2.

## 4 Quantile-Regression-based Counterfactual Prediction: Practical Approaches

Building on the theoretical results in Section 3, we develop a practical framework for counterfactual prediction based on quantile regression. This approach removes the need to estimate the underlying structural causal model or the noise distribution. We first formulate counterfactual prediction as a bi-level optimization problem that jointly infers the factual quantile level and its corresponding quantile function (Section 4.1). Next, we present a neural implementation (Section 4.2) that enables efficient estimation under general causal mechanisms and complex data distributions. Finally, Section 4.3 establishes the uniqueness and optimality of the proposed bi-level formulation, ensuring a well-defined counterfactual prediction, and analyzes its generalization behavior by deriving an upper bound on the prediction error.

### 4.1 Counterfactual Prediction under a Bi-Level Framework

Let $\{(x_i, y_i, z_i)\}_{i=1}^N$ denote observed realizations of the random variables $X$, $Y$, and $Z$. Given a specific factual observation $\langle x, y, z \rangle$, our goal is to predict the counterfactual outcome $Y_{x'}$ under the intervention $do(X = x')$.

We characterize counterfactual outcomes through conditional quantiles. For any quantile level $\tau \in (0, 1)$, the true conditional quantile function is defined as

$$\mu_\tau^\star(x, z) = \inf\{v \mid P(Y \le v \mid X = x, Z = z) \ge \tau\}. \tag{1}$$

Given the factual sample $\langle x, y, z \rangle$, the associated (instance-specific) quantile level is

$$\tau^\star := P(Y \le y \mid X = x, Z = z). \tag{2}$$

According to Lemma 1, under the intervention $do(X = x')$, the counterfactual outcome satisfies

$$Y_{x'} = \mu_{\tau^\star}^\star(x', z). \tag{3}$$

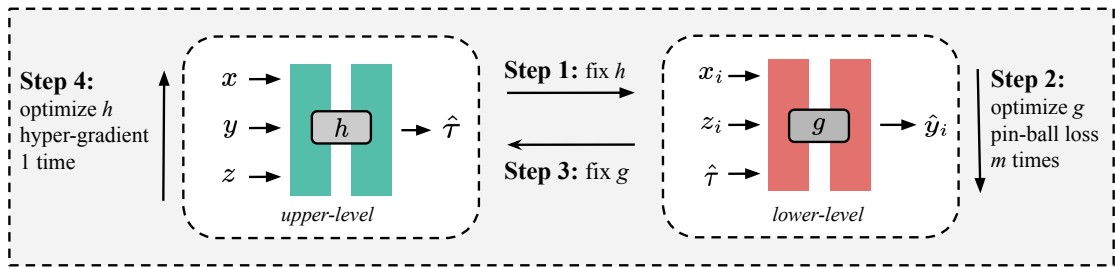

Figure 3: Illustration of the bi-level training procedure. The lower-level network $g$ learns the conditional quantile function by minimizing the empirical pinball loss (Step 2) while keeping $h$ fixed (Step 1). The upper-level network $h$ predicts the quantile level $\hat{\tau}$ and updates its parameters once per iteration via the hypergradient of the upper-level loss (Step 4), with $g$ held fixed (Step 3). This alternating optimization jointly minimizes $\widehat{\mathcal{L}}_{\text{lower}}$ and $\widehat{\mathcal{L}}_{\text{upper}}$, realizing the neural bi-level framework of Equation (8).

In practice, both $\tau^\star$ and $\mu_\tau^\star$ are unknown. A naive strategy is to estimate them sequentially: first estimating the cumulative distribution function (CDF) to obtain $\tau^\star$, and then evaluating the learned quantile function. However, accurate estimation of the conditional CDF $P(Y \leq y \mid X = x, Z = z)$ requires a large number of samples in the local neighborhood of $(x, z)$. In high-dimensional settings, this data sparsity leads to unstable estimates of $\tau^\star$ (see Fig. 5).

To overcome this, we jointly infer the quantile level and the conditional quantile function via a bi-level optimization framework. Let $\mathcal{F}$ denote a class of measurable functions used to parameterize conditional quantile regressors (e.g., neural networks).

**Lower Level (Distribution Learning):** For a fixed quantile level $\tau \in (0, 1)$, the conditional quantile function $\mu_\tau \in \mathcal{F}$ is learned by minimizing the population quantile regression loss:

$$\mu_\tau \in \arg\min_{q \in \mathcal{F}} \ \mathcal{L}_{\text{lower}}(q; \tau), \tag{4}$$

where

$$\mathcal{L}_{\text{lower}}(q; \tau) = \mathbb{E}_{(X,Y,Z)}\big[\ell_\tau\big(Y - q(X, Z)\big)\big]. \tag{5}$$

Here, $\ell_\tau$ denotes the pinball loss (Takeuchi et al., 2006):

$$\ell_\tau(\xi) = \begin{cases} \tau\,\xi, & \xi \geq 0, \\ (\tau - 1)\,\xi, & \xi < 0. \end{cases} \tag{6}$$

**Upper Level (Instance Identification):** The latent quantile level $\tau$ for the *specific* factual observation $\langle x, y, z \rangle$ is selected by minimizing the discrepancy between the predicted quantile and the observed outcome:

$$\mathcal{L}_{\text{upper}}(\tau; x, y, z) = \phi\big(y, \mu_\tau(x, z)\big), \tag{7}$$

where $\phi$ is a distance metric (e.g., squared error $\phi(y, \hat{y}) = (y - \hat{y})^2$).

Combining these, we formulate the prediction process for a given instance as the following bi-level optimization problem:

$$\min_{\tau \in (0,1)} \ \mathcal{L}_{\text{upper}}(\tau; x, y, z) \quad \text{s.t.} \quad \mu_\tau \in \arg\min_{q \in \mathcal{F}} \ \mathcal{L}_{\text{lower}}(q; \tau). \tag{8}$$

This formulation anchors the identification of $\tau$ to the globally learned function $\mu_\tau$, avoiding the instability associated with local estimation methods.

## 4.2 Neural Realization of the Bi-Level Framework

To operationalize the abstract objectives in Equation (8), we introduce a neural parameterization that explicitly models the bi-level structure. We employ two cooperating neural networks: a *quantile-level network*

$h_{\theta_h}$ (upper level) and a *conditional quantile network* $g_{\theta_g}$ (lower level). These networks are trained jointly to minimize the coupled objectives derived from the bi-level formulation.

**Neural parameterization.** The *quantile-level network* $h_{\theta_h}(x, y, z)$ acts as an encoder, mapping a factual observation to its latent quantile estimate $\hat{\tau} \in (0, 1)$. The *conditional quantile network* $g_{\theta_g}(x, z, \hat{\tau})$ acts as generator, outputting the predicted outcome value $\hat{\mu}_{\hat{\tau}}(x, z)$ for a given context and quantile level. The composition of these networks, $g(x, z, h(x, y, z))$, represents the model's reconstruction of the observed outcome $y$.

**Empirical lower-level optimization.** For a fixed set of quantile levels $\hat{\tau}$ generated by $h$, the lower-level network $g$ is optimized to minimize the empirical pinball loss. This encourages $g$ to accurately model the conditional distribution specified by $\hat{\tau}$:

$$\widehat{\mathcal{L}}_{\text{lower}}(\theta_g; \theta_h) = \frac{1}{N} \sum_{i=1}^{N} \ell_{\hat{\tau}_i}(y_i - g(x_i, z_i, \hat{\tau}_i)), \qquad \text{where } \hat{\tau}_i = h(x_i, y_i, z_i). \tag{9}$$

Here, $\ell_{\hat{\tau}}(\cdot)$ is the asymmetric pinball loss defined in Equation (5).

**Empirical upper-level optimization.** The upper-level network $h$ is updated to minimize the discrepancy between the observed outcome and the value predicted by the optimal lower-level model. The objective is given by:

$$\widehat{\mathcal{L}}_{\text{upper}}(\theta_h; \theta_g^{\star}) = \frac{1}{N} \sum_{i=1}^{N} \phi(y_i, g(x_i, z_i, \hat{\tau}_i)), \tag{10}$$

where $\phi(\cdot)$ is a discrepancy metric (e.g., $\ell_1$ or $\ell_2$ loss) and $\theta_g^{\star}(\theta_h)$ denotes the optimal lower-level parameters given the current encoder $h$. Crucially, updating $\theta_h$ requires computing the *hypergradient*, which accounts for how changes in $\hat{\tau}$ influence the optimal solution of the lower-level problem $g^{\star}$:

$$\nabla_{\theta_h} \widehat{\mathcal{L}}_{\text{upper}} = \frac{\partial \widehat{\mathcal{L}}_{\text{upper}}}{\partial \theta_h} + \frac{\partial \widehat{\mathcal{L}}_{\text{upper}}}{\partial \theta_g^{\star}} \cdot \frac{d\theta_g^{\star}}{d\theta_h}. \tag{11}$$

Computing the Jacobian $\frac{d\theta_g^{\star}}{d\theta_h}$ directly is computationally prohibitive. To address this, we employ SAMA (Choe et al., 2023a), which provides an efficient approximation for implicit gradient differentiation in bi-level optimization.

**Joint training.** As illustrated in Fig. 3, we employ an alternating optimization strategy. In each iteration, $h$ generates quantile estimates $\hat{\tau}$ for the batch. Keeping $h$ fixed, we update $g$ for $m$ steps to minimize the lower-level pinball loss Equation (9). Once $g$ is updated, we freeze it and update $h$ by one step using the hypergradient estimate from Equation (11). This alternating process ensures that $h$ learns to identify quantile levels that are consistent with the distributional shapes learned by $g$.

**Practical considerations regarding monotonicity.** Theoretically, our bi-level formulation relies solely on the pinball loss without incorporating external regularization. Consequently, it does not strictly prevent issues such as quantile crossing in highly flexible neural networks. We opt not to enforce strict monotonicity primarily to avoid the optimization complexity and reduced expressivity associated with rigid architectural constraints, such as those used in BGM (Nasr-Esfahany et al., 2023). Empirically, however, we observe that severe quantile crossing does not occur in our evaluated settings (see, for example, Fig. 7 and Fig. 4). Nevertheless, if a specific application requires strict adherence to the monotonicity assumption, one could readily extend our framework by incorporating explicit non-crossing penalties (e.g., Tagasovska & Lopez-Paz, 2019) or by utilizing inherently monotonic network architectures.

### 4.3 Theoretical Guarantees of the Bi-level Quantile Regression

Having introduced the bi-level framework, we now provide its theoretical foundations. We first establish that the proposed optimization is well-posed: under the monotonic SCM condition in Lemma 1, the bi-level

objective admits a *unique* global optimum corresponding to the true quantile level and conditional quantile value. We then analyze how the empirical estimators generalize to unseen counterfactual inputs.

**Population-level Analysis: Uniqueness and Identifiability of Counterfactual Quantile Estimation.**
We now establish that the proposed bi-level formulation uniquely identifies the instance-specific quantile level $\tau^\star$ together with the corresponding conditional quantile function value used for counterfactual prediction.

**Theorem 1.** *Under the assumptions of Lemma 1, and assuming that the true conditional quantile function is contained within the lower-level hypothesis class $\mathcal{F}$, the bi-level optimization problem in Equation* (8) *admits a unique global minimizer given by the pair* $(\hat{\tau}, \hat{\mu}_{\hat{\tau}})$*, where*

$$\hat{\tau} = P(Y \leq y \mid X = x, Z = z), \qquad \hat{\mu}_{\hat{\tau}}(x, z) = \inf\{v \mid P(Y \leq v \mid X = x, Z = z) \geq \hat{\tau}\}.$$

*Moreover, the counterfactual outcome under the intervention $do(X = x')$ is uniquely identified as*

$$Y_{x'} = \hat{\mu}_{\hat{\tau}}(x', z).$$

The proof is provided in Appendix D. Theorem 1 establishes that the proposed bi-level formulation is well posed and identifiable: it admits a unique global optimum corresponding to the true conditional quantile level and its associated quantile function. Consequently, the bi-level framework recovers the true counterfactual outcome.

**Remark on Theory vs. Implementation:** We note that Theorem 1 characterizes the global minima of the idealized, population-level bi-level objective. It guarantees that our formulation is mathematically sound: if the global minimum is reached, it yields the true counterfactual. However, it does not provide convergence guarantees for the empirical procedure detailed in Eqs. 9–11. In practice, we rely on parameterized neural networks, alternating optimization, and approximate hypergradients. While bridging this optimization gap for non-convex architectures remains an open challenge, our empirical results in Table 1 demonstrate that this approximate procedure performs well in our evaluated settings, indicating that the theoretical formulation translates effectively to practical implementation.

**Generalization Analysis** An important question is whether the learned regressor $\hat{\mu}_{\hat{\tau}}$ generalizes to unseen counterfactual inputs $(x', z)$ that differ from those observed during training. Formally, we aim to bound the expected excess risk:

$$\mathbb{E}_{X,Z}\left[\ell_{\hat{\tau}}\big(\mu_{\hat{\tau}}^\star(X, Z) - \hat{\mu}_{\hat{\tau}}(X, Z)\big)\right],$$

which measures the deviation between the learned conditional quantile and the population quantile. Below, we derive an upper bound on this error using the Rademacher complexity of the hypothesis class.

**Definition 2** (Rademacher complexity (Bartlett & Mendelson, 2002))**.** Let $\mathcal{F}$ be a hypothesis class mapping from $\mathcal{X}$ to $[0, b]$. Let $\{(x_i, z_i)\}_{i=1}^N$ be i.i.d. examples. Let $\{\sigma_i\}_{i=1}^N$ be independent Rademacher variables taking values in $\{-1, +1\}$ uniformly. The Rademacher complexity is defined as

$$\mathfrak{R}(\mathcal{F}) = \mathbb{E}_{X,Z,\sigma}\left[\sup_{\mu \in \mathcal{F}} \frac{1}{N} \sum_{i=1}^N \sigma_i \mu(x_i, z_i)\right]. \tag{12}$$

Our main theoretical result is as follows.

**Theorem 2** (Generalization Bound for the Conditional Quantile Regression Component)**.** *Let $(\hat{\tau}, \hat{\mu}_{\hat{\tau}})$ be the solution obtained from the optimization, where $\hat{\mu}_{\hat{\tau}} \in \mathcal{F}$. Assume the loss function $\ell_{\hat{\tau}}$ is upper bounded by $b$. Then, for any $\delta > 0$, with probability at least $1 - \delta$, we have*

$$\mathbb{E}_{X,Z}\left[\ell_{\hat{\tau}}\big(\mu_{\hat{\tau}}^\star(X, Z) - \hat{\mu}_{\hat{\tau}}(X, Z)\big)\right] \leq \frac{1}{N} \sum_{i=1}^N \ell_{\hat{\tau}}\big(\mu_{\hat{\tau}}^\star(x_i, z_i) - \hat{\mu}_{\hat{\tau}}(x_i, z_i)\big)$$

$$+ 4\mathfrak{R}(\mathcal{F}) + \frac{4b}{\sqrt{N}} + b\sqrt{\frac{\log(1/\delta)}{2N}}. \tag{13}$$

Figure 4: Counterfactual estimation examples for the five common structural causal model (SCM) classes introduced in Section 3. For each SCM, we vary $X$ across intervention values $x'$ and compare the predicted outcomes with the ground-truth counterfactuals $Y_{X=x'}$. The proposed bi-level method accurately recovers the true counterfactual trajectory in each case.

The proof is provided in Appendix E. The Rademacher complexity has been widely used to derive generalization error bounds in the statistical machine learning community (Mohri et al., 2018). If $\mathcal{F}$ is a Reproducing Kernel Hilbert Space (RKHS) and the hypotheses are bounded, then without requiring strong assumptions, $\mathfrak{R}(\mathcal{F}) \leq O(\sqrt{1/N})$ (Bartlett & Mendelson, 2002).

**Discussions and Implications** The upper bound of the expected excess risk $\mathbb{E}_{X,Z}[\ell_{\hat{\tau}}(\mu_{\hat{\tau}}^{\star}(X, Z) - \hat{\mu}_{\hat{\tau}}(X, Z))]$ for the learned conditional quantile regressor heavily relies on two key factors: (i) the empirical error term $\frac{1}{N} \sum_{i=1}^{N} \ell_{\hat{\tau}}(\mu_{\hat{\tau}}^{\star}(x_i, z_i) - \hat{\mu}_{\hat{\tau}}(x_i, z_i))$, which is effectively minimized by optimizing the model on the factual training samples, and (ii) the number of training samples $N$. Given a fixed number of training samples, our theoretical results imply that the generalization error of this lower-level quantile estimation is strictly bounded. However, we note that Theorem 2 provides a guarantee for the conditional quantile regression component at the estimated quantile level, rather than a finite-sample guarantee for the full, end-to-end counterfactual predictor. Establishing a strict global bound is highly non-trivial, as it would require systematically accounting for the estimation error of the upper-level quantile level $\tau$, potential model misspecification, and the optimization error induced by the alternating training procedure. While such a comprehensive theoretical analysis remains an important direction for future work, our framework proves highly effective in practice. Complementing the theoretical insights of Theorem 2, we empirically demonstrate in Section 5.3 that our method achieves significantly better performance than strong baselines given the same sample size. In summary, our framework provides solid theoretical grounding for the conditional quantile regression component, which successfully translates into robust empirical performance for counterfactual prediction.

## 5 Experimental Results

In this section, we evaluate the proposed method along three axes. First, we analyze the learned quantiles under diverse structural models. Second, we conduct comprehensive comparisons with state-of-the-art approaches across tabular and image datasets. Finally, we study sample efficiency and assess robustness to violations of key assumptions, including approximate monotonicity (high-dimensional outputs) and latent confounding.

### 5.1 Analysis of Quantile Learning

We begin with controlled experiments to examine how the proposed bi-level formulation facilitates quantile learning and counterfactual prediction. Specifically, we examine (i) whether the bi-level structure is necessary, (ii) how it improves stability over Monte Carlo estimation, and (iii) whether it recovers the true quantile levels and counterfactual trajectories, thereby empirically supporting our theoretical guarantees.

**Experimental setup.** We construct synthetic datasets where the ground-truth quantile levels $\tau$ are analytically available, enabling direct comparison between learned and true values. Specifically, we instantiate the five structural causal models (SCMs) described in Section 3: (1) a linear model, $Y = X + Z + E$; (2) a nonlinear additive noise model, $Y = \sin(2\pi X + Z) + E$; (3) a nonlinear multiplicative noise model, $Y = \exp(X - Z + 0.5) \cdot E$; (4) a post-nonlinear model, $Y = \exp(\sin(\pi X + Z) + E)$; and (5) a heteroscedastic

Table 1: Root Mean Squared Error (RMSE) performance for counterfactual prediction on tabular datasets. Because true counterfactual outcomes are never observed during training, models are trained only on factual samples, and evaluation is performed on both the training and test splits to assess generalization. RankPrev (Wu et al., 2025) is designed specifically for binary interested variables and is therefore evaluated only in IHDP and Dis-Dose.

| Method | IHDP | | Cont-Dose | | Dis-Dose | |
|---|---|---|---|---|---|---|
| | Train | Test | Train | Test | Train | Test |
| DeepSCM (Pawlowski et al., 2020) | $2.37 \pm 2.$ | $2.98 \pm 4.$ | $0.38 \pm .0$ | $0.40 \pm .0$ | $0.33 \pm .0$ | $0.36 \pm .0$ |
| CFQP-T (De Brouwer, 2022) | $1.81 \pm .1$ | $1.80 \pm .1$ | $0.19 \pm .0$ | $0.19 \pm .0$ | $0.22 \pm .0$ | $0.22 \pm .0$ |
| CFQP-U (De Brouwer, 2022) | $1.40 \pm .1$ | $1.30 \pm .0$ | $0.19 \pm .0$ | $0.18 \pm .0$ | $0.22 \pm .0$ | $0.22 \pm .0$ |
| BGM (Nasr-Esfahany et al., 2023) | $4.35 \pm .4$ | $4.89 \pm .5$ | $0.31 \pm .0$ | $0.39 \pm .1$ | $0.27 \pm .0$ | $0.29 \pm .0$ |
| DCM (Chao et al., 2023) | $2.56 \pm 2.$ | $2.76 \pm 2.$ | $0.19 \pm .0$ | $0.16 \pm .0$ | $0.28 \pm .0$ | $0.29 \pm .0$ |
| RankPrev (Wu et al., 2025) | $2.54 \pm 2.$ | $2.48 \pm 2.$ | - | - | $0.22 \pm .0$ | $0.22 \pm .0$ |
| Discretized Quantile Search | $1.37 \pm .2$ | $1.35 \pm .2$ | $0.08 \pm .0$ | $0.08 \pm .0$ | $0.27 \pm .0$ | $0.28 \pm .0$ |
| Ours | $\mathbf{1.29 \pm .3}$ | $\mathbf{1.23 \pm .2}$ | $\mathbf{0.06 \pm .0}$ | $\mathbf{0.06 \pm .0}$ | $\mathbf{0.20 \pm .0}$ | $\mathbf{0.20 \pm .0}$ |

model, $Y = \exp(-5X + Z) + \exp(X + Z - 0.5) \cdot E$. In all cases, the true quantile relation satisfies $F^{-1}(\tau) = E \Rightarrow \tau = F(E)$, where $F$ is the CDF of the noise variable $E$. We set $P(X) = P(Z) = \mathcal{U}[0,1]$ and $E \sim \mathcal{N}(0,1)$, generating 100,000 samples per SCM. For evaluation, we fix $X = 0.5$, $Z = 0.5$, and $E = 0.5$ to obtain the corresponding $Y$, where the ground-truth quantile is $\tau^\star = \Phi(0.5) \approx 0.691$.

**(1) Necessity and advantage of the bi-level formulation.** A naïve approach to counterfactual estimation is to fit a family of quantile regressors at pre-specified quantile levels (e.g., 0.1, 0.5, 0.9) and then search exhaustively for the one that best matches the observed outcome. This two-stage procedure is computationally expensive, sensitive to grid resolution, and unable to exploit information from the factual observation $(x, z, y)$. Moreover, standard quantile regressors, especially linear ones, fail to capture nonlinear relationships. In contrast, our bi-level formulation jointly optimizes the quantile level $\tau$ and the quantile function $\mu_\tau(x, z)$, directly aligning $\mu_\tau(x, z)$ with the observed $y$. As shown in Fig. 4, this coupling allows the model to efficiently locate the correct quantile $\tau^\star$ without exhaustive search and to recover the true counterfactual trajectories across all SCMs. We further confirm this necessity through an ablation study (Table 1), where we replace our bi-level formulation with a discrete grid search ($\tau \in [0, 1]$ at 0.01 intervals) over our trained neural quantile regressor. Although this discretized baseline is highly competitive, our full bi-level approach significantly outperforms it. This highlights that optimizing the quantile estimation and counterfactual prediction jointly, rather than sequentially, is crucial for maximizing performance.

**(2) Stability compared with Monte Carlo estimation.** Monte Carlo (MC) estimation approximates the quantile level via $\hat{\tau} = \frac{1}{m} \sum_{i=1}^{m} \mathbb{I}(y_i \leq y \mid x_i = x, z_i = z)$, where $m$ is the number of samples near $(x, z)$. In practice, the number of such samples is small, especially in high-dimensional or discrete settings, making MC estimates noisy and unstable. Fig. 5 shows that MC quantiles fluctuate substantially across runs, whereas our bi-level approach produces consistent and accurate estimates. By coupling the learning of $\tau$ and $\mu_\tau$, the model leverages information from the entire population rather than relying on local sample counts, thereby avoiding the sparsity and variance problems of Monte Carlo estimation. This coupling yields significantly more stable and data-efficient inference.

**(3) Verification of the Unique Global Minimizer.** Fig. 4 shows that the predicted counterfactual trajectories $\hat{Y}_{x'}$ align almost perfectly with the ground truth across all causal models. Moreover, as shown in Fig. 5, the inferred quantile levels $\hat{\tau}$ closely coincide with the analytical values $\tau^\star = F(Y \mid X, Z)$, where $F$ is the cumulative distribution function, indicating that our method successfully recovers the true conditional quantiles. These findings provide strong empirical support for Theorem 1, confirming that the proposed bi-level optimization admits a unique global minimizer and produces reliable counterfactual predictions.

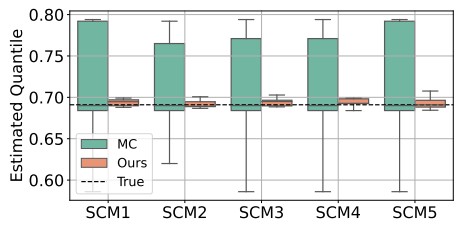

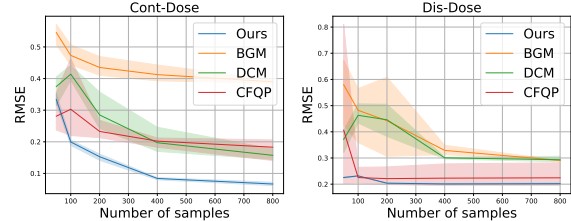

Figure 5: Box plots of the learned quantiles.

Figure 6: The study of sample efficiency.

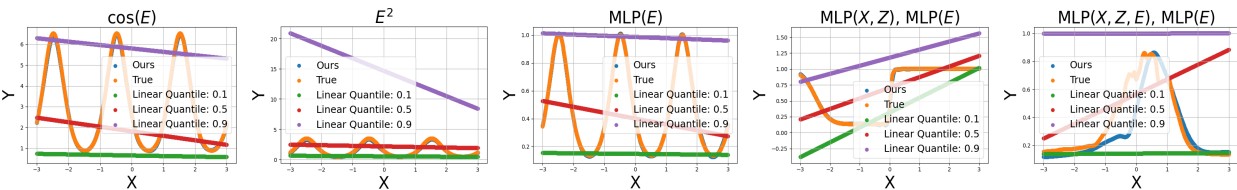

Figure 7: Analysis on the monotonicity assumption. (a)$Y = \exp(\cos(\pi X + 3Z) + \cos(E))$; (b)$Y = \exp(\cos(\pi X + 3Z) + E^2)$; (c)$Y = \exp(\cos(\pi X + 3Z) + \mathrm{MLP}(E))$; (d)$Y = \exp(\mathrm{MLP}(X, Z) + \mathrm{MLP}(E))$; (e)$Y = \exp(\mathrm{MLP}(X, Z, E) + \mathrm{MLP}(E))$. Although $Y$ is not monotonic w.r.t $E$, the counterfactual outcome is still identifiable since we may have $r(E) = E^2, \cos(E), \mathrm{MLP}(E)$ and $Y$ is monotonic w.r.t $r(E)$. Empirically, we can see that the counterfactual predictions are very close to the ground truth in the first four cases. As for the last case (e), it exhibits deviation from the truth since the monotonicity assumption of function $f$ is violated and the counterfactual outcome may not be identifiable.

## 5.2 Comparisons with State-of-the-Art Approaches

Because real-world counterfactual outcomes are unobservable, we evaluate our method on simulated datasets that satisfy the monotonicity assumption in Lemma 1. In the *Cont-Dose* and *Dis-Dose* datasets, the covariate $Z$ represents patient age, $X$ denotes the treatment dose (continuous and binary, respectively), and $Y$ is the outcome generated by $Y = X^2 + 5\log(5X) - Z^{0.5} + 2E$. Each dataset contains 800 training and 200 test samples. We further include the semi-synthetic *IHDP* dataset (Hill, 2011) for a more realistic evaluation.

We compare our approach against several state-of-the-art baselines: DeepSCM (Pawlowski et al., 2020), CFQP (De Brouwer, 2022), BGM (Nasr-Esfahany et al., 2023), DCM (Chao et al., 2023), and RankPrev (Wu et al., 2025). For CFQP, we report results for both the Transformer-based variant (CFQP-T) and the U-Net-based variant (CFQP-U). All baselines are implemented using their public code and recommended hyperparameters. As shown in Table 1, our method consistently achieves the lowest RMSE across all datasets, demonstrating the effectiveness of the proposed quantile-regression-based bi-level framework for counterfactual prediction.

## 5.3 Generalization Bound and Sample Efficiency

Theorem 2 establishes that the expected generalization error of our estimator is bounded by the empirical risk and the number of training samples. To empirically verify this result, we conduct an ablation study varying the training set size on the *Cont-Dose* and *Dis-Dose* datasets. We train each method with $N \in \{50, 100, 200, 400, 800\}$ samples and report the average RMSE over multiple runs in Fig. 6. The results show that baseline methods, BGM (Nasr-Esfahany et al., 2023) and CFQP (De Brouwer, 2022), are highly sensitive to sample size. In particular, CFQP exhibits large variance when $N < 200$, likely due to its clustering-based structure, which further reduces the effective number of samples per cluster. In contrast, our bi-level method consistently achieves the lowest error with minimal variance, confirming the theoretical generalization bound and demonstrating superior sample efficiency.

Table 2: The RMSE performance for counterfactual prediction on image transformation datasets.

| Method | Rotation-MNIST | | Thick-Omniglot | |
|---|---|---|---|---|
| | Train | Test | Train | Test |
| DeepSCM | $6.61 \pm .1$ | $6.55 \pm .1$ | $11.31 \pm .1$ | $11.46 \pm .1$ |
| CFQP-T | $2.82 \pm .1$ | $2.80 \pm .0$ | $4.36 \pm .0$ | $4.23 \pm .0$ |
| CFQP-U | $1.79 \pm .0$ | $1.78 \pm .0$ | $3.28 \pm .1$ | $3.30 \pm .1$ |
| BGM | $7.53 \pm .2$ | $7.52 \pm .1$ | $12.07 \pm .2$ | $12.27 \pm .1$ |
| DCM | $5.30 \pm .0$ | $5.23 \pm .0$ | $8.80 \pm .0$ | $9.01 \pm .1$ |
| RankPrev | - | - | - | - |
| Ours | $\mathbf{1.54 \pm .1}$ | $\mathbf{1.52 \pm .1}$ | $\mathbf{2.60 \pm .0}$ | $\mathbf{2.96 \pm .1}$ |

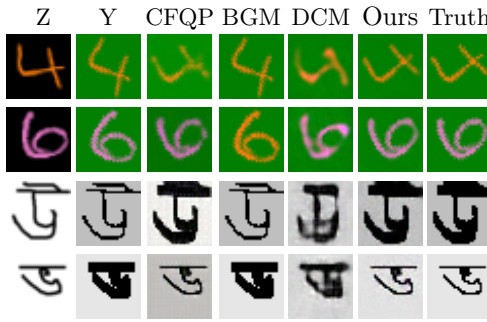

Figure 8: Example results on image transformation dataset.

## 5.4 Assumption Analysis

An essential assumption of our method is the existence of a latent function $r(E)$ such that the outcome $Y$ is strictly monotonic with respect to $r(E)$. Because $r(\cdot)$ can take arbitrary forms, this assumption encompasses a broad class of structural causal mechanisms and is generally well supported by the strong empirical performance of our approach. Nevertheless, monotonicity may not hold exactly in all real-world scenarios. To evaluate the robustness of our method beyond this assumption, we consider three complementary settings: (i) tabular data with non-monotonic structural functions, (ii) image-based tasks where the high-dimensional nature of $Y$ introduces mild departures from strict monotonicity, and (iii) cases involving latent confounders that induce correlations among variables potentially violating the monotonicity assumption.

**When $Y$ is not monotonic in $E$.** We begin by examining the five causal models illustrated in Fig. 7. In the first four models, although $Y$ is not strictly monotonic in $E$, reparameterizations such as $r(E) = E^2$, $\cos(E)$, or MLP$(E)$ make $Y$ monotonic in $r(E)$, thereby satisfying the identifiability conditions in Theorem 1. The resulting counterfactual predictions align closely with the ground truth, providing strong empirical support for our theoretical analysis. In the fifth model, the monotonicity assumption fails because the structural function combines nonlinear transformations of both $(X, Z)$ and $E$ in a non-separable way, preventing any monotone mapping of $E$ that preserves identifiability. Consequently, our predictions exhibit small but noticeable deviations from the ground truth, as expected. Handling fully non-monotonic generative processes remains an open and important direction for future research.

**When $Y$ is high-dimensional.** We next evaluate cases with high-dimensional outcomes $Y$, which better reflect real-world data where relationships are complex yet remain largely monotonic in latent factors. To study this setting, we construct two image-based datasets: *Rotation-MNIST* and *Thick-Omniglot*. In *Rotation-MNIST*, $Z$ denotes the original MNIST digits (LeCun et al., 2010), $X$ specifies the rotation angle, and the noise term corresponds to RGB perturbations of the rotated images, following the generative process $Y =$ Rotate$(X, Z) + E$. In *Thick-Omniglot*, $Z$ represents Omniglot characters (Lake et al., 2019), $X$ controls stroke thickness, and the noise term accounts for variations in image darkness, with $Y = $ AdjustThickness$(Z, X) \times E$. Although $Y$ is high-dimensional, we construct datasets such that its pixel intensities form partially ordered sets, ensuring that quantiles remain well defined. The quantile regression objective (pinball loss) is applied element-wise across all pixels, and then averaged to form the final lower-level loss. Additional dataset details are provided in Appendix F.

Table 2 reports the RMSE on these image-based counterfactual prediction tasks. Even though strict monotonicity may not hold, our method achieves the lowest RMSE on both datasets, outperforming all baselines. This result indicates that the proposed bi-level framework generalizes effectively to complex, high-dimensional outcomes, a property essential for real-world applications. In contrast, BGM (Nasr-Esfahany et al., 2023) performs worse, as its conditional spline flow lacks sufficient expressiveness for modeling intricate image transformations.

Fig. 8 presents qualitative examples of counterfactual image generation. CFQP (De Brouwer, 2022) often fails to preserve background color or structural consistency in the predicted images, suggesting difficulties in correctly identifying the noise realization and leading to unreliable counterfactuals. Our method, by contrast, accurately models the transformations and preserves global appearance, further demonstrating its robustness and practical identifiability.

**When latent confounders exist.** Following prior work (Nasr-Esfahany et al., 2023; Pawlowski et al., 2020; Chao et al., 2023), our theoretical analysis assumes no latent confounders. To test robustness when this assumption is violated, we consider four cases with an unobserved confounder $C \sim \text{Unif}[-0.5, 0.5]$ and independent noise $E \sim \mathcal{N}(0, 1)$. (1) In the no-confounder case, data are generated by $Y = XZ + E$. (2) In the $X \leftarrow C \rightarrow Z$ case, $X = E_X + C$, $Z = E_Z + C$, and $Y = XZ + E$. (3) In the $Z \leftarrow C \rightarrow Y$ case, $Z = E_Z + C$ and $Y = XZ + E + C$. (4) In the $X \leftarrow C \rightarrow Y$ case, $X = E_X + C$ and $Y = XZ + E + C$.

Table 3: Counterfactual prediction performance under latent confounder.

| Case | RMSE↓ |
|---|---|
| No $C$ | $0.06 \pm 0.0$ |
| $X \leftarrow C \rightarrow Z$ | $0.07 \pm 0.0$ |
| $Z \leftarrow C \rightarrow Y$ | $0.07 \pm 0.0$ |
| $X \leftarrow C \rightarrow Y$ | $0.22 \pm 0.1$ |

As shown in Table 3, our method performs well in the first three cases but exhibits a performance drop when the confounder jointly affects both $X$ and $Y$. This difference arises from how the effective noise $r(E)$ interacts with $X$ across the four settings. In the first two cases, $r(E) = E$, and the monotonicity assumption holds exactly. In the third and fourth cases, $r(E) = E + C$. When the confounder $C$ influences $Z$ and $Y$ but not $X$, as in the third case, $r(E)$ remains independent of $X$. As a result, intervening on $X$ does not alter the properties of $r(E)$, and the model maintains stable performance. In contrast, when $C$ affects both $X$ and $Y$, as in the fourth case, $r(E)$ becomes dependent on $X$, violating the monotonicity assumption and leading to higher RMSE.

## 6 Conclusion

Counterfactual prediction remains a fundamental challenge due to the unobservability of counterfactual outcomes in real-world data. In this paper, we develop a neural bi-level optimization framework that jointly estimates the quantile function and the corresponding quantile level. We prove that the resulting population-level optimization admits a unique global minimizer, providing a theoretical foundation for reliable counterfactual predictions, and further provide a generalization bound for the empirical estimator. Comprehensive experiments on simulated and semi-synthetic datasets demonstrate that our method consistently outperforms state-of-the-art baselines, indicating its practical effectiveness and robustness in these evaluated settings.

## Acknowledgements

We would like to acknowledge the support from NSF Award No. 2229881, AI Institute for Societal Decision Making (AI-SDM), the National Institutes of Health (NIH) under Contract R01HL159805, and grants from Quris AI, Florin Court Capital, MBZUAI-WIS Joint Program, and the Al Deira Causal Education project.

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

## A  More Related Work

Individual treatment effect (ITE) estimation is also closely related to the counterfactual inference problem. A major difference is that ITE focuses on the effect of treatment and the predictions of both factual and counterfactual outcomes on unseen samples. GANITE (Yoon et al., 2018) first learns a counterfactual generator with GAN by matching the joint distribution of observed covariate and outcome variables, and then it generates a dataset by feeding different treatment values and random noises and learns an ITE generator to predict the factual and counterfactual outcomes. CFRNet (Johansson et al., 2016) models ITE as a domain adaptation problem where there is a distribution shift between effects under different treatment and matches the marginal distributions of representations under different treatments in the representation space. ABCEI (Du et al., 2021) proposes to use adversarial learning to balance the representations from treatment and control groups. CBRE (Zhou et al., 2022) proposes to use cycle consistency to preserve the semantics of the representations from two groups. Based on GANITE, SCIGAN (Bica et al., 2020) further proposes a hierarchical discriminator to learn the counterfactual generator when interventions are continuous, e.g., the dosage of the treatment. SITE (Yao et al., 2018) uses propensity score to select positive and negative pairs and proposes to minimize the middle point distance to preserve the relationships in the representation space. Based on SITE, CITE (Li & Yao, 2022) employs contrastive learning to preserve the relationships. BV-NICE (Lu et al., 2020b) models the generation process as a latent variable model where a confounder causes the treatment, covariate, and outcomes and addresses the covariate imbalance with adversarial training. The goal of Zhou et al. (2021b) is to estimate uncertainty intervals by learning two networks in an adversarial manner, where one is to estimate CDF and the other is to estimate the quantile. Later, Zhou et al. (2021a) extend (Zhou et al., 2021b) to the ITE task. Xie et al. (2020); Powell (2020) propose ways to estimate the quantile treatment effects unlike the average treatment effect. The quantile treatment effect is measured on all samples with same value of the quantile. On the contrary, our method learns different quantile for different individuals and uses the quantile to represent the property of the conditional distribution.

## B  The Noise Distribution

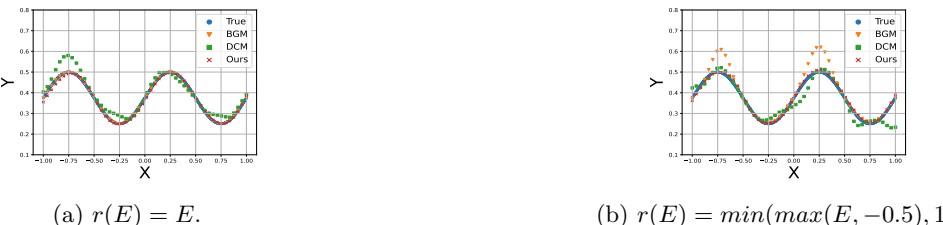

(a) $r(E) = E$.                    (b) $r(E) = min(max(E, -0.5), 1)$

Figure 9: Comparisons of the counterfactual predictions when traversing $X$ under two different forms of $r(E)$. Our method, based on Lemma 1, demonstrates resilience to various forms of $r(E)$.

For instance, consider the following two scenarios: in the first scenario, let $Y = \frac{1}{4}(Z + \sin(2\pi X)r(E) + 2r(E))$, $E$ satisfies a Gaussian distribution, and $r(E) = E$, while in the second scenario, $r(E) = \min(\max(E, -0.5), 1)$. For a particular sample $\langle X = 0.5, Z = 0.5 \rangle$ (the unknown noise $E = 0.5$), we have $Y = 1.5$ for both scenarios. Now we estimate the counterfactual outcome when $X$ had been set to different values, with results illustrated in Figure 9. We can see our approach, which relies on Lemma 1, remains unaffected by the form of $r(E)$, since in both scenarios, it relies on $P(Y \leq 1.5 | X = 0.5, Z = 0.5)$. However, BGM and DCM exhibit vastly different performances in the two scenarios because they assume Gaussian noise terms, while it is impossible to recover a noise term that is both Gaussian and satisfies invertibility when $r(E) = \min(\max(E, -0.5), 1)$.

## C  Proof of Lemma 1

*Proof.* Note $Y = f(X, Z, E)$ can be equivalently represented as $f_1(f_2(X, Z), r(E))$, and we denote $r(E)$ by $\tilde{E}$. We know that without further restrictions on the function class of $f$, the causal model $f$ and the probabilistic distribution $p(\tilde{E})$ are not identifiable (Zhang et al., 2015). Denote by $f^i$ and $p^i(\tilde{E})$ as one solution, and we

will see that the counterfactual outcome actually does not depend on the index $i$; that is, it is independent of which $f^i$ and $P^i(\tilde{E}_{t+1})$ we choose. Given observed evidence $(X = x, Y = y, Z = z)$, because $f^i$ is strictly monotonic in $\tilde{E}^i$, we can determine its value $\tilde{e}^i$, with $\tilde{e}^i = f^{i}_{x,z}{}^{-1}(y)$. Then, we can determine the value of the cumulative distribution function of $\tilde{E}^i$ at $\tilde{e}^i$, denoted by $\tau^i$.

Without loss of generality, we first show the case where $f^i$ is strictly increasing w.r.t. $\tilde{E}^i$. Because $f$ is strictly increasing in $\tilde{E}$ and $y = f^i(x, z, \tilde{e}^i)$, $y$ is the $\tau^i$-th quantile of $P(Y|X = x, Z = z)$. Then it is obvious that since $y$ and $P(Y|X = x, Z = z)$ are determined, the value of $\tau^i$ is independent of the index $i$, that is, it is identifiable. Thus, below, we will use $\tau$, instead of $\tau^i$.

Since $r(E) \perp\!\!\!\perp (X; Z)$, when doing interventions on $X$, the value $\tilde{e}^i$ will not change, as well as $e^i$. Hence, the counterfactual outcome $Y_{X=x'}|X = x, Y = y, Z = z$ can be calculated as $f^i(X = x', Z = z, E = e^i)$, and such equivalence can be directly derived following Pearl's three-step procedure. Because $\tilde{e}^i$ does not change after the intervention, the counterfactual outcome $Y_{X=x'}|X = x, Y = y, Z = z$ is the $\tau$-quantile of the conditional distribution $P(Y|X = x', Z = z)$. This quantile exists and it depends only on the conditional distribution $P(Y|X = x', Z = z)$, but not the chosen function $f^i$ and $P^i(\tilde{E})$.

Therefore, the counterfactual outcome $Y_{X=x'}|X = x, Y = y, Z = z$ corresponds to the $\tau$-th quantile of the conditional distribution $P(Y|X = x', Z = z)$, where $Y = y$ represents the $\tau$-th quantile of $P(Y|X = x, Z = z)$. $\qquad\square$

## D Proof of Theorem 1

*Proof.* The proof proceeds in three steps: (1) analyzing the optimality of the lower-level optimization under the realizability assumption, (2) establishing the uniqueness of the upper-level solution, and (3) confirming the identifiability of the counterfactual outcome.

**Step 1: Optimality of the Lower Level.** Consider the lower-level optimization problem defined in Equation (4). For a fixed $\tau \in (0, 1)$, the objective is to minimize the expected pinball loss over the hypothesis class $\mathcal{F}$:

$$\min_{q \in \mathcal{F}} \mathbb{E}(X, Y, Z) \left[ \ell\tau(Y - q(X, Z)) \right]. \tag{14}$$

It is a standard result in quantile regression theory (Takeuchi et al., 2006) that the unconstrained population minimizer of the pinball loss is the true conditional $\tau$-th quantile function of $Y$ given $X$ and $Z$. Because we assume that this true conditional quantile function, denoted $\mu_\tau^\star$, is contained within the lower-level hypothesis class $\mathcal{F}$, the obtained solution $\hat{\mu}_\tau \in \mathcal{F}$ successfully achieves this theoretical minimum and satisfies

$$\hat{\mu}\tau(x, z) = \mu\tau^\star(x, z) = \inf v \mid P(Y \le v \mid X = x, Z = z) \ge \tau \tag{15}$$

almost surely. Thus, for any input $\tau$, the lower level recovers the true conditional quantile function.

**Step 2: Existence and Uniqueness of the Upper Level Solution.** Substituting the optimal lower-level solution $\mu_\tau^\star$ into the upper-level objective Equation (7), the optimization problem requires finding the root of

$$y = \mu_\tau^\star(x, z). \tag{16}$$

By definition, the conditional quantile function $\mu_\tau^\star$ is the inverse of the strictly increasing conditional cumulative distribution function (CDF), denoted $F_{Y|X,Z}$. Thus, we can rewrite Equation (16) as

$$y = F_{Y|X,Z}^{-1}(\tau). \tag{17}$$

Applying $F_{Y|X,Z}$ to both sides recovers the unique quantile level:

$$\tau = F_{Y|X,Z}(y) \equiv P(Y \le y \mid X = x, Z = z). \tag{18}$$

The existence of this solution is guaranteed by the Intermediate Value Theorem (due to the continuity of $F_{Y|X,Z}$ implied by the smoothness of $f$), and its uniqueness is guaranteed by the strict monotonicity of $F_{Y|X,Z}$.

**Step 3: Counterfactual Identifiability.** Having identified the unique quantile level $\hat{\tau} = \tau^{\star}$ and the conditional quantile function $\hat{\mu}_{\hat{\tau}} = \mu_{\tau^{\star}}^{\star}$, we address the counterfactual prediction. According to Lemma 1, under the intervention $do(X = x')$, the noise term (and thus the quantile level) remains invariant. Therefore, the true counterfactual outcome is given by the $\tau^{\star}$-th quantile of the interventional distribution:

$$Y_{x'} = \mu_{\tau^{\star}}^{\star}(x', z). \tag{19}$$

Since the bi-level framework recovers $\hat{\tau} = \tau^{\star}$ and $\hat{\mu} = \mu^{\star}$, the predicted value is

$$\hat{Y}x' = \hat{\mu}\hat{\tau}(x', z) = \mu_{\tau^{\star}}^{\star}(x', z) = Y_{x'}. \tag{20}$$

This confirms that the counterfactual outcome is uniquely identified by the proposed framework. $\square$

# E   Proof of Theorem 2

**Motivation.**   Theorem 2 establishes that the generalization error of our estimator is bounded by the empirical loss on the training data. This implies that minimizing the empirical pinball loss effectively learns a quantile estimator that remains accurate on the population. As a consequence of this bound and the identifiability result in Theorem 1, our method is capable of performing reliable counterfactual prediction using only finite factual training samples.

**Notation.**   In this section, to simplify notation, we use $f$ to represent the conditional quantile function $\mu$. Furthermore, we omit $z$ in the function arguments (writing $f(x)$ instead of $f(x, z)$), as the concatenation of $z$ and $x$ can be treated as a single input vector without affecting the derivation of the bound.

## E.1   Preliminaries

We first restate the standard generalization error bound derived based on Rademacher complexity (Bartlett & Mendelson, 2002; Mohri et al., 2018).

**Lemma 2** (Generalization Bound via Rademacher Complexity). *Let $\mathcal{H}$ be a hypothesis class of functions mapping from $\mathcal{X}$ to $[0, b]$. Let $S = \{x_i\}_{i=1}^{N}$ be i.i.d. training samples. Then, for any $\delta > 0$, with probability at least $1 - \delta$, the following holds for all $h \in \mathcal{H}$:*

$$\mathbb{E}[h(x)] \le \frac{1}{N} \sum_{i=1}^{N} h(x_i) + 2\mathfrak{R}_N(\mathcal{H}) + b\sqrt{\frac{\log(1/\delta)}{2N}}. \tag{21}$$

Inspired by this result, we derive the specific bound for our counterfactual quantile regression problem.

## E.2   Proof Derivation

Let $\mathcal{L}_{\mathfrak{T},\mathcal{F}}$ denote the class of loss functions induced by our model class $\mathcal{F}$ and parameter space $\mathfrak{T} = (0, 1)$, defined as:

$$\mathcal{L}_{\mathfrak{T},\mathcal{F}} = \{(x, y) \mapsto \ell_\tau(f^{\star}(x) - f(x)) \mid \tau \in \mathfrak{T}, f \in \mathcal{F}\}.$$

Applying Lemma 2 to this loss class yields the following intermediate bound:

**Theorem 3.** *Let $(\hat{\tau}, \hat{f}_{\hat{\tau}})$ be the optimization solution. Let the loss function $\ell_\tau$ be upper bounded by $b$. Then, for any $\delta > 0$, with probability at least $1 - \delta$:*

$$\mathbb{E}_{(x,y)}[\ell_{\hat{\tau}}(f^{*}(x) - \hat{f}(x))] \le \frac{1}{N} \sum_{i=1}^{N} \ell_{\hat{\tau}}(f^{*}(x_i) - \hat{f}(x_i))$$

$$+ 2\mathfrak{R}_N(\mathcal{L}_{\mathfrak{T},\mathcal{F}}) + b\sqrt{\frac{\log(1/\delta)}{2N}}, \tag{22}$$

*where the Rademacher complexity of the loss class is*

$$\mathfrak{R}_N(\mathcal{L}_{\mathfrak{T},\mathcal{F}}) = \mathbb{E}_{S,\sigma}\left[\sup_{\tau \in \mathfrak{T}, f \in \mathcal{F}} \frac{1}{N}\sum_{i=1}^{N}\sigma_i \ell_\tau(f^*(x_i) - f(x_i))\right]. \tag{23}$$

To complete the proof of Theorem 2, we must bound the complexity term $\mathfrak{R}_N(\mathcal{L}_{\mathfrak{T},\mathcal{F}})$ in terms of the model complexity $\mathfrak{R}_N(\mathcal{F})$.

**Lemma 3.** *The Rademacher complexity of the loss class is bounded by:*

$$\mathfrak{R}_N(\mathcal{L}_{\mathfrak{T},\mathcal{F}}) \le 2\mathfrak{R}_N(\mathcal{F}) + \frac{2b}{\sqrt{N}}. \tag{24}$$

*Proof.* Recall the definition of the pinball loss for a residual $u = f^*(x) - f(x)$:

$$\ell_\tau(u) = \tau u - \mathbb{I}_{\{u<0\}}u,$$

where $\mathbb{I}_{\{\cdot\}}$ is the indicator function. Substituting this into the definition of Rademacher complexity:

$$\mathfrak{R}_N(\mathcal{L}_{\mathfrak{T},\mathcal{F}}) = \mathbb{E}_{S,\sigma}\left[\sup_{\tau,f}\frac{1}{N}\sum_{i=1}^{N}\sigma_i\left(\tau(f^*(x_i) - f(x_i)) - \mathbb{I}_{\{f^*(x_i)-f(x_i)<0\}}(f^*(x_i) - f(x_i))\right)\right]$$

$$\le \underbrace{\mathbb{E}_{S,\sigma}\left[\sup_{\tau,f}\tau\frac{1}{N}\sum_{i=1}^{N}\sigma_i(f^*(x_i) - f(x_i))\right]}_{T_1}$$

$$+ \underbrace{\mathbb{E}_{S,\sigma}\left[\sup_{f}\frac{1}{N}\sum_{i=1}^{N}\sigma_i\left(-\mathbb{I}_{\{f^*(x_i)-f(x_i)<0\}}(f^*(x_i) - f(x_i))\right)\right]}_{T_2}. \tag{25}$$

**Bounding $T_1$:** Since $\tau \in (0,1)$, the function $g(u) = \tau u$ is a contraction (Lipschitz constant $\le 1$). By the properties of suprema and Talagrand's contraction lemma, this term is bounded by the complexity of the residuals:

$$T_1 \le \mathbb{E}_{S,\sigma}\left[\sup_{f \in \mathcal{F}}\frac{1}{N}\sum_{i=1}^{N}\sigma_i(f^*(x_i) - f(x_i))\right]. \tag{26}$$

**Bounding $T_2$:** The term $-\mathbb{I}_{\{u<0\}}u$ is equivalent to $\max(0, -u) = \text{ReLU}(-u)$. Since the ReLU function is 1-Lipschitz, we again apply Talagrand's contraction lemma:

$$T_2 \le \mathbb{E}_{S,\sigma}\left[\sup_{f \in \mathcal{F}}\frac{1}{N}\sum_{i=1}^{N}\sigma_i(f^*(x_i) - f(x_i))\right]. \tag{27}$$

**Combining Terms:** Combining $T_1$ and $T_2$, we have:

$$\mathfrak{R}_N(\mathcal{L}_{\mathfrak{T},\mathcal{F}}) \le 2\mathbb{E}_{S,\sigma}\left[\sup_{f \in \mathcal{F}}\frac{1}{N}\sum_{i=1}^{N}\sigma_i(f^*(x_i) - f(x_i))\right]$$

$$= 2\mathbb{E}_{S,\sigma}\left[\sup_{f \in \mathcal{F}}\frac{1}{N}\sum_{i=1}^{N}\sigma_i(-f(x_i))\right] + 2\mathbb{E}_{S,\sigma}\left[\frac{1}{N}\sum_{i=1}^{N}\sigma_i f^*(x_i)\right]$$

$$= 2\mathfrak{R}_N(\mathcal{F}) + 2\mathbb{E}_{S,\sigma}\left[\frac{1}{N}\sum_{i=1}^{N}\sigma_i f^*(x_i)\right], \tag{28}$$

where we used the property $\mathfrak{R}_N(\{-f \mid f \in \mathcal{F}\}) = \mathfrak{R}_N(\mathcal{F})$.

Finally, we bound the constant term involving the oracle $f^*$. Using Jensen's inequality and the fact that $f^*$ is bounded by $b$:

$$
\begin{aligned}
\mathbb{E}_{S,\sigma}\left[\frac{1}{N}\sum_{i=1}^{N}\sigma_i f^*(x_i)\right] &\leq \frac{1}{N}\left(\mathbb{E}_{S,\sigma}\left[\left(\sum_{i=1}^{N}\sigma_i f^*(x_i)\right)^2\right]\right)^{1/2} \\
&= \frac{1}{N}\left(\mathbb{E}_S \sum_{i=1}^{N}(f^*(x_i))^2\right)^{1/2} \quad (\text{since } \mathbb{E}[\sigma_i\sigma_j]=0 \text{ for } i\neq j) \\
&\leq \frac{1}{N}\sqrt{Nb^2} = \frac{b}{\sqrt{N}}.
\end{aligned}
\tag{29}
$$

Substituting this back yields the result: $\mathfrak{R}_N(\mathcal{L}_{\mathfrak{T},\mathcal{F}}) \leq 2\mathfrak{R}_N(\mathcal{F}) + \frac{2b}{\sqrt{N}}$. $\qquad\square$

**Proof of Theorem 2.** Substituting the result of Lemma 3 into Equation (22) in Theorem 3, we obtain:

$$
\begin{aligned}
\mathbb{E}[\ell_{\hat{\tau}}(f^* - \hat{f})] &\leq \widehat{\mathbb{E}}[\ell_{\hat{\tau}}(f^* - \hat{f})] + 2\left(2\mathfrak{R}_N(\mathcal{F}) + \frac{2b}{\sqrt{N}}\right) + b\sqrt{\frac{\log(1/\delta)}{2N}} \\
&= \frac{1}{N}\sum_{i=1}^{N}\ell_{\hat{\tau}}(f^*(x_i) - \hat{f}(x_i)) + 4\mathfrak{R}_N(\mathcal{F}) + \frac{4b}{\sqrt{N}} + b\sqrt{\frac{\log(1/\delta)}{2N}}.
\end{aligned}
\tag{30}
$$

This completes the proof. $\qquad\square$

## F  Datasets

**Datasets**. Since our main result is based on the monotonicity assumption and the observations in counterfactual scenarios are lacking in real-world, we create following datasets.

**Rotation-MNIST**. We use MNIST images (LeCun et al., 2010) as $Z$ and rotation angles as $X$. Then we randomly sample a noise value from $U[0,1]$ and adjust the rotated images by adding values to the RGB channels. Therefore, the final pixel values are strictly monotonic w.r.t the noise value when conditioning on $X$ and $Z$. The training set consists of 60000 images while the testing set consists of 10000 images. The dimension of $Z$ is $3 \times 32 \times 32 = 3072$ and the dimension of $X$ is 1, which determines the rotation angles within [-45,45]. The dimension of $Y$ is also $3 \times 32 \times 32$. In counterfactual inference, we are given a sample $X, Z, Y$, the question asks what had the images would be if the rotation angle is set to another value. In other words, we aim to rotate the digits in $Y$ while avoiding any other pixel value changes.

**Thick-Omniglot**. We use Omniglot images (Lake et al., 2019) as $Z$ and the thickness of characters as $X$. Then we randomly sample a noise value from $U[0,1]$ and adjust the background darkness by multiplying the thicken characters pixel values. Therefore, the final pixel values are strictly monotonic w.r.t the noise value when conditioning on $X$ and $Z$. The number of images is 19280 and We split the dataset as 80/20. We also resize the images to 32×32. In this dataset, our goal is to adjust the thickness by the new value of $X$ while preserving the brightness in $Y$.

**Cont-Dose**. We generate this dataset by setting $X$ as the treatment and $Z$ as the age of patient and the outcome $Y$ denotes the effect. $X$ takes value from [0,2] with space 0.1 uniformly. The number of training samples is 800 and the number of testing samples is 200.

**Dis-Dose**. We generate this dataset by setting $X$ as the treatment and $Z$ as the age of patient and the outcome $Y$ denotes the effect. $X$ takes value from $\{0,1\}$. The numbers of training samples is 800 and the number of testing samples is 200.

We also use the semi-simulated dataset IHDP (Hill, 2011). It contains 100 splits. Each split consists of 675 training and 92 testing samples. The dimension of $Z$ is 25 and the dimension of $X$ is 1 and the dimension of output $Y$ is 1.

## G   Implementation

**All codes are provided in the supplementary materials. Readers may refer to the code for more implementation details.**

We use betty (Choe et al., 2023b) as our bi-level optimization library. Then we define the lower level loss as the pin-ball loss with the quantile passed from the upper level. After optimizing the lower level net $g$ with 30 iterations, we fix the network $g$ and optimize the upper-level network $h$ on the regression loss (We adopt MSE loss).

For toy datasets and cont-Dose dataset, we build the upper-level network as 3 layers and 1 output linear layer. The network $h$ takes as input the factual observations $X$, $Z$ and $Y$. Then we concatenate the three variables and feed into the model. The first layer is a linear layer transforming the input into a hidden-dimension (200) and apply a SiLU activation function. Then we use 2 residual blocks (with layernorm) and finally feed the output into the linear layer with output size as 1. The final activation is Sigmoid since we need to use $h$ to represent quantile, which is within range $[0,1]$. As for the lower-level network $g$, it is mainly trained to get a regression at quantile $\tau$, which is the output of network $h$. We first project the input $X, Z, \tau$ into same dimensions respectively, then we feed into a linear layer to match the dimension of $Y$.

For Dis-Dose and IHDP, $X$ is binary. Therefore, we use two networks with same architecture as $h$. We feed the inputs $X, Z, Y$ into different sub-network according to the value $X$. We find that a shallower network is better, therefore, we do not use residual block here. For the network $h$, we use 3-layers (one input layer and 2 residual blocks).

For image transformation datasets, we use the convolutional neural network to parameterize $h$ and $g$. The hidden dimension of $h$ is 32 while the hidden dim of $g$ is 128. We downsample the images with Conv(4,2,1) and final conv(4,1,0) to get the quantile estimations. In network $g$, we use a symmetric network where the upsampling network mimics the downsampling process with ConvTranspose(4,2,1).

We use $n_i$ interested samples in training network $g$. For each interested sample, we have one quantile $\tau$. Then we train the network $g$ with $n_t$ samples for each $\tau$. We set $n_i = 256, n_t = 64$ for most cases. As for image datasets, we have to lower the values to reduce memory. So we use $n_i = 128, n_t = 32$. In the upper-level problem, we use 64 (128) samples to get the reconstruction loss. We use Adam optimizer with lr=2e-3 for images while 1e-3 for the rest of them.

We choose the hyper-parameters based on the values on some toy datasets since we can always generate them. Then we apply the best hyper-parameters to the formal dataset, Dis-Dose, Cont-Dose, IHDP, Thick-Ominiglot and Rotation-MNIST. During training, we use the reconstruction error on the training dataset as metric to select the models.

## H   Confounding

In the main paper, we considered a simple case where $Y = X + Z + E$, where $Z \sim U[0,1], X \sim U[0,1]$ and $e \sim N(0,1)$. We generate the latent confounder as $C \sim U[-0.5, 0.5]$. Then we add the confounder to its children in different scenarios. For example, when we have $Z \leftarrow C \rightarrow Y$, we add $Z = Z + C, Y = Y + C$. According to the table in the main paper, the performances drop on the scenario $X \leftarrow C \rightarrow Y$ in this case, which is kindly expected since we do not assume the existence of latent confounder and $C$ influences $X$ and $Y$ now. (the code of data generation and training is provided in the supp.)

## I   More Results on Image Transformation Datset

**Learned Quantiles**. Since we sample the noise values uniformly from $[0,1]$. The target quantile $P(Y \leq y | X = x, Z = z)$ is the CDF of the noise. In other words, if the noise is $a$, the target quantile should be $a$. We present the examples of Thick-Omniglot in Fig. 11. Our estimated quantiles are not affected by the thickness changes between the input $Z$ and the output $Y$. The learned quantiles are very close to the ground

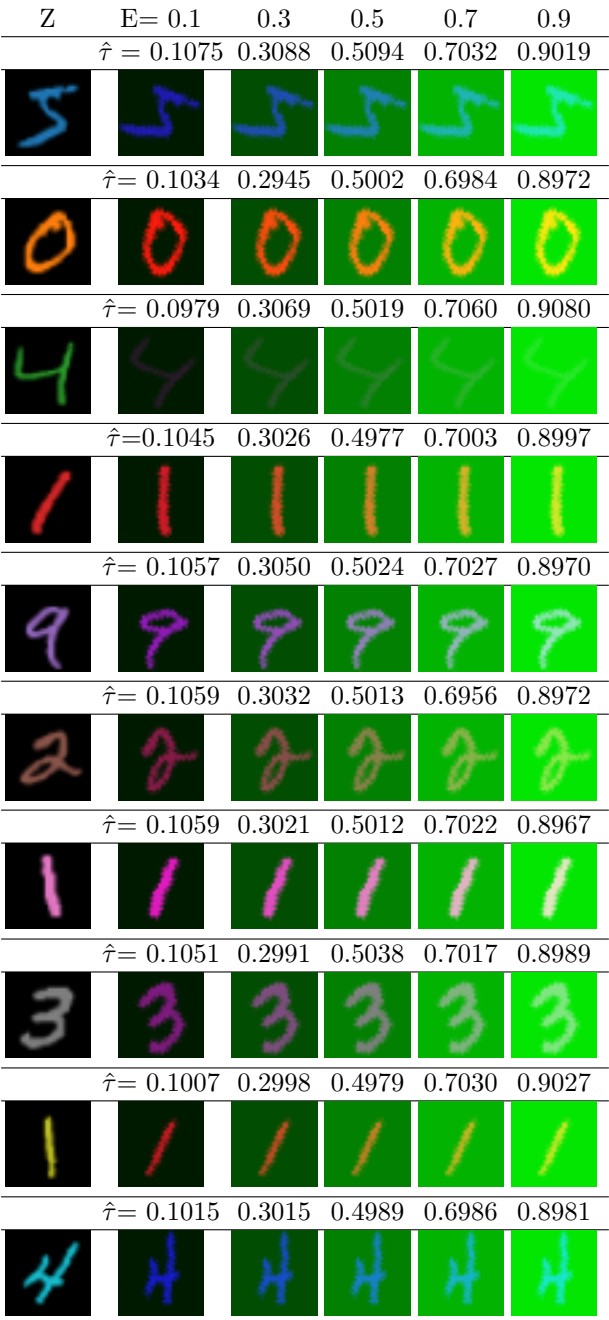

Figure 10: Examples of learned quantiles on Rotation-MNIST dataset. The learned quantiles $\hat{\tau}$ are very close to the true quantiles, i.e., 0.1, 0.3,0.5,0.7 and 0.9.

truth, further demonstrating the effectiveness of our method. We also provide examples of Rotation-MNIST in Fig. 10. Our method is able to recover the quantiles of the true noises accurately.

**Counterfactual Prediction Results** We present more visual examples of Rotation-MNIST in Fig. 12 and 13. We can see that the CFQP (De Brouwer, 2022) learns the rotation when changing the value of $X$. However, the rotations results is unsatisfactory. In particular, the background color of the rotated digits are different from the ground truth $Y_{X=x'}$ and the $Y$. This is unwanted because $X$ only affects the rotation. It indicates that CFQP fails to capture the uniqueness (noise) of the sample and leads to unnecessary changes.

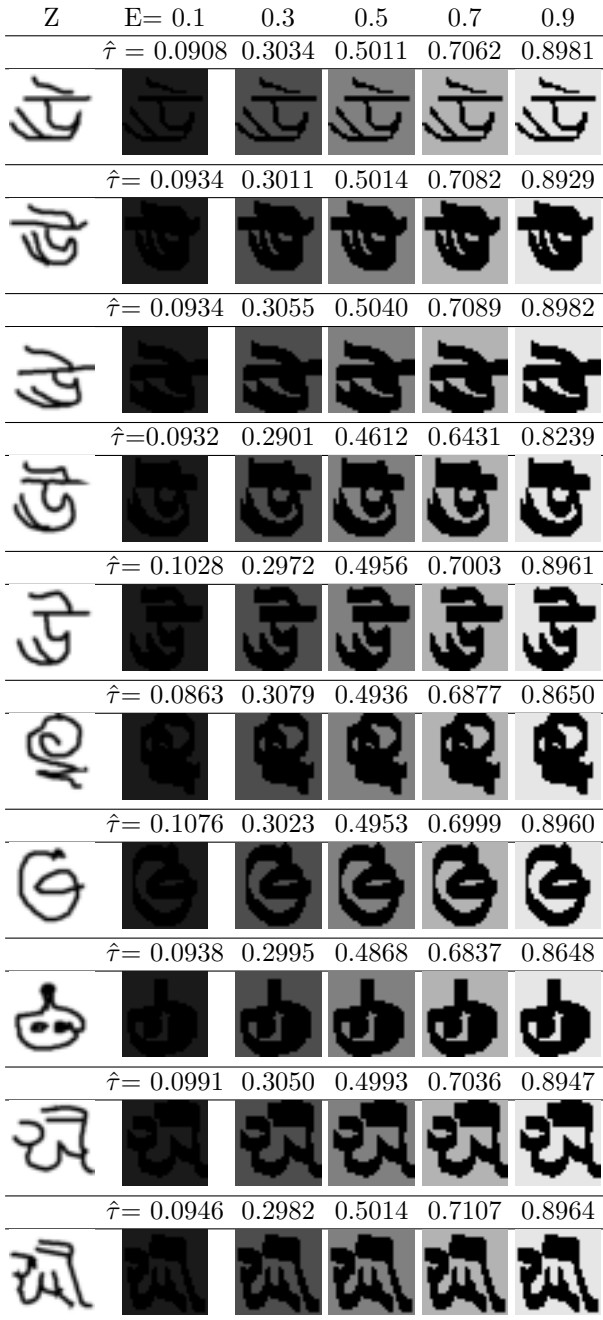

Figure 11: Examples of learned quantiles on Thick-Omniglot dataset. The learned quantiles $\hat{\tau}$ are very close to the true quantiles, i.e., 0.1, 0.3, 0.5, 0.7 and 0.9. Each row displays the images after adjusting thickness by $X$ and the brightness with noise value. The noise determines the brightness of the images. From the $Z$ and the output image, our method is not affected by the changing thickness between $Z$ and $Y$ and able to capture the brightness changes. The accurate estimation shows the effectiveness of our approach even when $Z$ and $Y$ are high-dimensional.

As for BGM (Nasr-Esfahany et al., 2023), the images are almost identical to the $Y$. The reason is that the BGM employs conditional spline flow to mimic the strictly monotonic generation process. However, it is known that the expressive power of flow, especially such strictly monotonic flow, are limited compared to the unconstrained neural networks. It fails to transform the high-dimensional input $Y$ into gaussian noise and therefore, fails to utilize the conditions ($X$ and $Z$). As a consequence, when we change the value of $X$, BGM ignores the condition and generates almost identical images to the input $Y$. DCM (Chao et al., 2023) models the generation process with conditional diffusion model and recovers the noise by inverting the diffusion process. However, since the true noise is usually not Gaussian, the transformed noise may not be meaningful to counterfactual inference. Therefore, the counterfactual predictions are not accurate. In contrast to above baseline methods, our approach learns the correct rotation as well as preserving the correct color in the factual observation $Y$.

We also present examples of Thick-Omniglot in Fig. 14. The noises values determines the darkness of the image. We observe that CFQP (De Brouwer, 2022) fails to preserve the darkness of the image $Y$. Sometimes the predictions are darker or much brighter. DCM (Chao et al., 2023) struggles to adjust the thickness of the images. In contrast, our method learns to preserve the brightness in the factual observation $Y$ while changing the thickness of the digits.

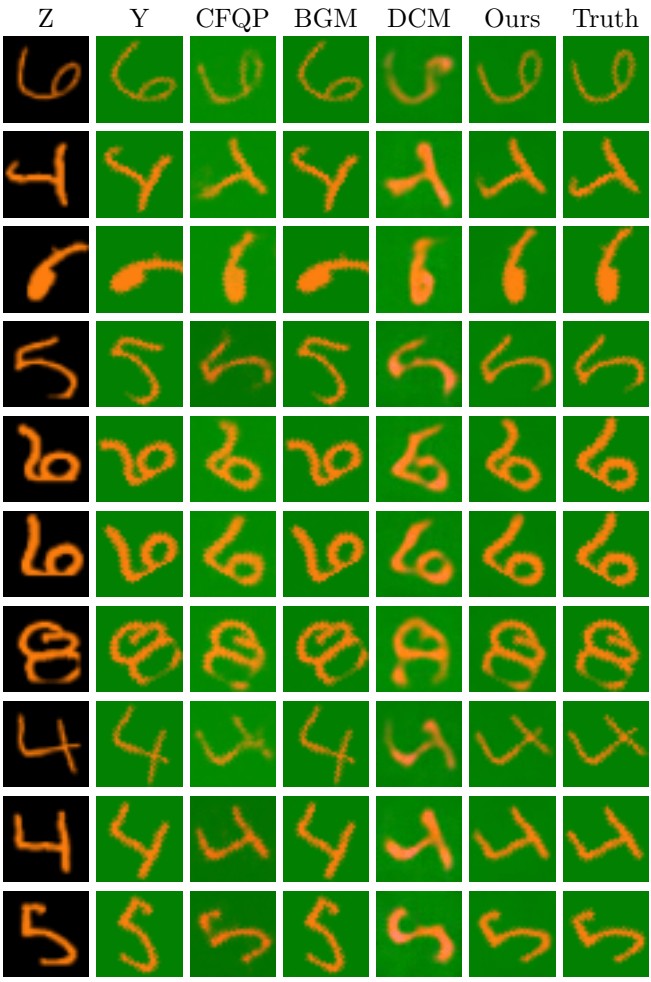

Figure 12: Examples of counterfactual predictions on Rotation-MNIST.

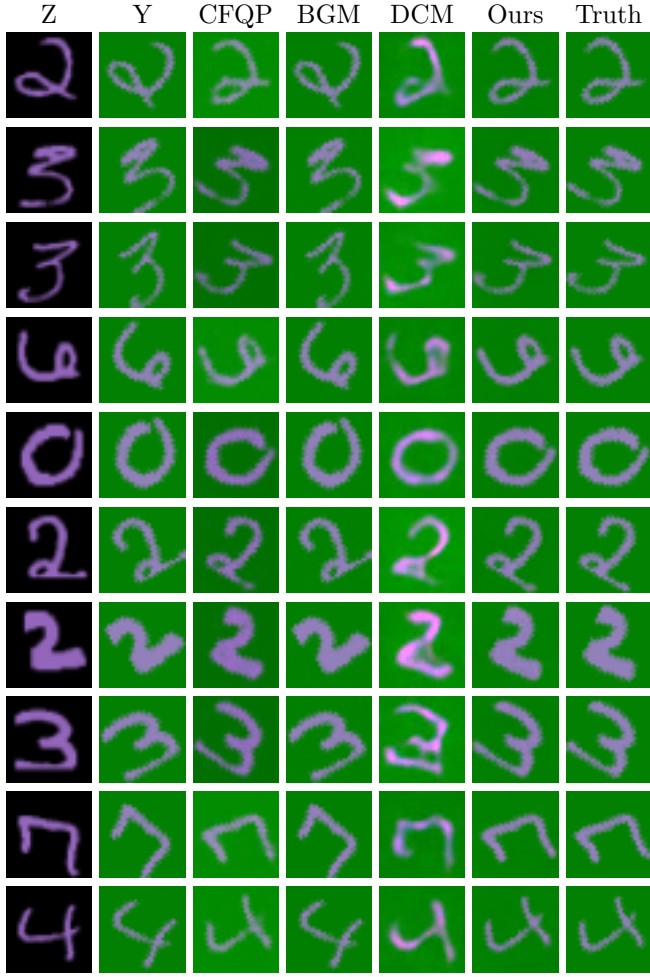

Figure 13: Examples of counterfactual predictions on Rotation-MNIST. CFQP fails to preserve the brightness of image $Y$ in their counterfactual predictions. BGM fails to transform the image $Y$ given the limited expressive power of the used spline flow. DCM generates unrealistic digits.

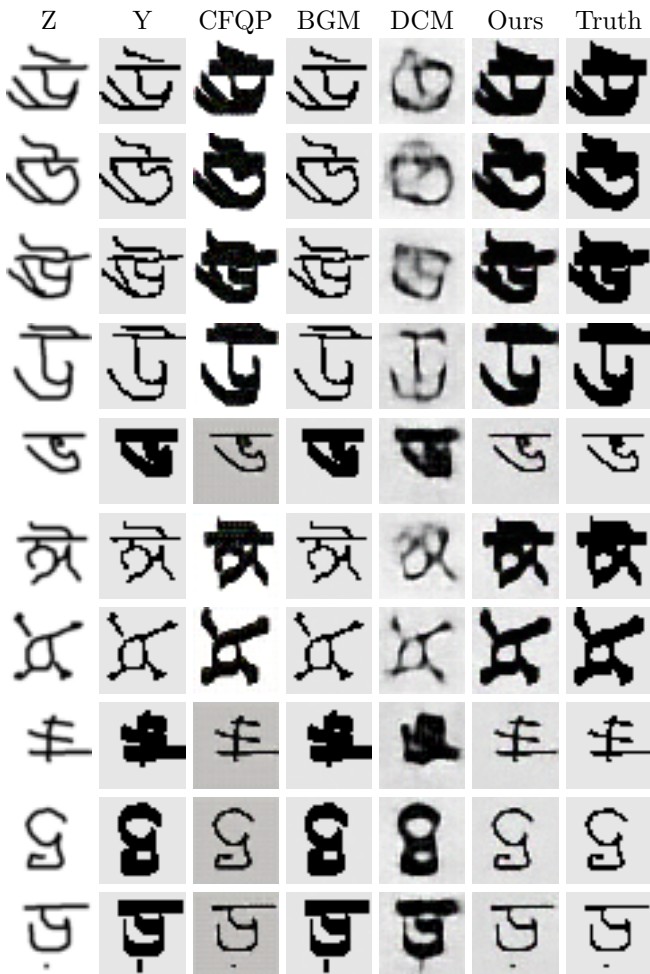

Figure 14: Examples of counterfactual predictions on Thick-Omniglot dataset.

