# OpenReview forum: "Advancing Counterfactual Prediction through Nonlinear Quantile Regression"
_TMLR — Accepted by TMLR_

### Review · Reviewer_CMZ5 · 2026-03-30

**Summary Of Contributions:**

The paper introduces a method for individual-level counterfactual prediction based on quantile regression, exploiting the rank preservation of quantile levels under intervention. While prior work has established connections between quantile regression and population-level causal inference, this paper is the first (to the best of my knowledge) to use quantile regression for individual-level counterfactual prediction.

The authors relax assumptions made in prior work by requiring only monotonicity in $r(E)$ rather than in the raw noise $E$ itself. As a result no distributional assumptions on $E$ or functional requirements on $r$ are needed, but the individual quantile level $\tau$ becomes the only identifiable quantity. Since $f$ is monotone in $r(E)$ and $r(E)$ is independent of $X$, $\tau$ is preserved under intervention. This is formalized in Lemma 1.

This setup reduces the problem of counterfactual prediction to learning the conditional quantile function and inferring the quantile level that matches the observed outcome. The authors address this by proposing a bi-level optimization scheme that exploits the fact that each subproblem is significantly more tractable when the other quantity is known/fixed. The proposed method alternates between fixing $\tau$ to learn the conditional quantile function by minimizing the pinball loss and fixing the quantile function and updating $\tau$ by taking a single gradient step based on the reconstruction error.

The authors then establish two theoretical guarantees. Theorem 1 proves that the bi-level optimization admits a unique global minimizer, corresponding exactly to the true $\tau$ and its associated conditional quantile value. Theorem 2 bounds the generalization error of the learned quantile estimator and gives finite sample performance guaranties.

**Additional Comments:**

The paper clearly states the assumptions on the model and empirically verifies what happens if these are violated. However, I believe that grounding the technical statement in a brief illustrative example of a real-world setting would go a long way for readers outside of causal inference. For instance, in a clinical trial comparing two weight loss drugs, the assumption implies that a patient who responds poorly to drug A would also respond poorly to drug B, i.e. their relative rank in the outcome distribution is preserved regardless of treatment. This is plausible when both drugs operate through the same broad mechanism, but fails when drugs work through fundamentally different pathways that interact differently with individual patient characteristics. Such an example would make it immediately clear to non-expert readers in which scenarios the assumption is reasonable and where it breaks. Currently this is left implicit in the technical discussion.

**Audience:**

Yes

**Audience Explanation:**

Yes, this paper meaningfully contributes to the field of causal inference. Moreover, despite its technical depth, the paper is written in a pedagogical way that makes its methodology and findings accessible to readers with only partial knowledge of causal inference.

**Claims And Evidence:**

Yes

**Claims Explanation:**

The claim made by the authors seem accurate and well supported by the presented evidence. (I have not checked the proofs and additional materials in the appendix.)

My only issue is that the authors seem to somewhat overstate the novelty of using quantile regression. Searching for connections of quantile regression and causal inference, I discovered the work from Chernozhukov & Hansen [1], who use quantile regression and rank invariance (as a direct assumption) on a population level. Suprisingly this work is not cited at all.
The authors' work is different from the mentioned work in multiple and significant way, but claim that they "establish a novel connection between counterfactual prediction and quantile regression", While technically correct, seems overstated knowing that the general connection between quantile regression estimation under intervention has been made previously. The authors' work extends this connection to the individual-level counterfactual setting, which is a meaningfully harder problem, but the paper and abstract do not acknowledge this prior work.

[1] Chernozhukov, V., & Hansen, C. (2005). An IV model of quantile treatment effects. Econometrica

**Requested Changes:**

I believe the authors should cite Chernozhukov & Hansen [1] and make sure that there are no other holes in the review of relevant literature, especially from the econometrics side of causal inference.

[1] Chernozhukov, V., & Hansen, C. (2005). An IV model of quantile treatment effects. Econometrica

---

> ### Author Response · Authors · 2026-05-05
> **Response to Reviewer CMZ5**
>
> We sincerely appreciate your kind recognition of the technical novelty of our work and the quality of its presentation. Your encouraging comments are greatly appreciated and have been very motivating for us. Below, we provide detailed responses to your questions. Please do not hesitate to let us know if you have any further questions or suggestions.
>
> ### Connection to Chernozhukov & Hansen (2005) and claims
>
> We thank the reviewer for this insightful comment and for pointing out the connection to Chernozhukov & Hansen (2005) [1]. We carefully reviewed this foundational work and completely agree with your assessment. We deeply appreciate your recognition that "the authors' work is different from the mentioned work in multiple and significant ways." Indeed, while [1] shares the core rank-invariance principle, it operates within the Instrumental Variable (IV) framework to resolve unobserved confounding, whereas we do not assume access to an instrument in our standard counterfactual prediction setting. Furthermore, classic econometric estimation methods for these models face scalability challenges in our context. As detailed in their follow-up work [2], solving the IV quantile regression often requires an inverse-inference procedure involving a grid search over the parameter space, which becomes computationally prohibitive for high-dimensional neural architectures.
>
>
>
>
> In light of your constructive suggestions, we have made the following revisions to our draft to properly credit this prior work and tone down our claims of novelty regarding the foundational connection:
>
>
> 1. In the Abstract and Section 3, we replaced "We establish a novel connection between counterfactual prediction and quantile regression, showing that..." with the following to more precisely describe our contribution:
>
> > "We show that counterfactual prediction can be reframed as an extended bi-level quantile regression problem."
>
>
> 2. In the Introduction Section, we added following to clarify that our contribution lies in operationalizing this quantile-preserving for individual-level prediction via a neural bi-level framework.::
>
> > While this theoretical insight is intuitive, effectively operationalizing it efficiently remains an open challenge. Although similar rank-preserving principles have been shared by previous works (Chernozhukov & Hansen,2005; Plečko & Meinshausen, 2020; Charpentier et al., 2023; De Lara et al., 2021; Machado et al., 2025; Wu et al., 2025), these methods face significant practical limitations. Specifically, they often assume access to instrumental variable or need to perform grid search over the parameter space (Chernozhukov & Hansen, 2005; Chernozhukov et al., 2017), the underlying density functions of the data-generating mechanism (Plečko& Meinshausen, 2020), rely on inaccurate high-dimensional density estimation (Wu et al., 2025) (see Table.1),or utilize optimal transport formulations where “implementation is challenging (except in the Gaussian case), and usually hard to interpret” (Machado et al., 2025).
>
> 3. In Related Work Section, we add:
> > Chernozhukov & Hansen (2005); Chernozhukov et al. (2017) establish rank invariance to identify quantile treatment effects using instrumental variables and explicit structural equations. However, their methodology relies on an inverse-inference procedure that typically requires a grid search over the parameter space, which is computationally prohibitive in high-dimensional settings. In contrast, we propose a direct neural bi-level optimization framework. Our approach learns quantile levels directly from observational data without specifying a structural model, yielding precise, point-valued individual counterfactual predictions via scalar quantile preservation.
>
> 4. In the discussion after Lemma1, we added:
> > Other methods also characterize counterfactual outcomes with quantiles, but often require access to the underlying data-generating densities (Plečko & Meinshausen, 2020) or rely on instrumental variables (Chernozhukov & Hansen, 2005; Chernozhukov et al., 2017)
>
>
> [1] Chernozhukov, V., & Hansen, C. (2005). An IV model of quantile treatment effects. Econometrica
>
> [2] Chernozhukov, Victor, Christian Hansen, and Kaspar Wüthrich. "Instrumental variable quantile regression." Handbook of quantile regression (2017): 119-143.

---

> > ### Author Response · Authors · 2026-05-05
> > **Response to Reviewer CMZ5 - Continued**
> >
> > ___
> >
> > ### Add an illustrative example to explain the monotonicity assumption
> >
> > We thank the reviewer for recognizing the clarity of our technical explanation and the empirical validation of the assumption. We completely agree that providing a more illustrative example significantly improves accessibility, especially for readers less familiar with causal inference.
> >
> > Following your excellent suggestion, we have added a concrete example immediately after Lemma 1 (highlighted in red in the revised manuscript) to clarify when the assumption is likely to hold or fail. The added text reads:
> >
> > > To provide intuition for the monotonicity assumption, consider a clinical setting comparing two weight-loss drugs. If both drugs act through a similar mechanism (e.g., appetite suppression), then patients who lose relatively less weight under drug A will also tend to lose relatively less weight under drug B. In this case, each patient’s position in the outcome distribution (their quantile level) is preserved across treatments, which is consistent with our assumption.
> >
> > > In contrast, if the two drugs operate through different mechanisms (e.g., one suppresses appetite while the other increases metabolic rate), patients may respond differently across treatments. A patient who responds poorly to one drug could respond well to the other, leading to changes in their relative position in the outcome distribution. In this case, the quantile level is not preserved, and the assumption underlying our method is violated."
> >
> >
> > We thank you for your suggestion regarding this important reference and for proposing the addition of this intuitive example. These improvements have made our manuscript clearer and more accurate, and have helped broaden its accessibility to a wider audience.

---

### Review · Reviewer_k5Rv · 2026-04-03

**Summary Of Contributions:**

This paper proposes a quantile-based approach to counterfactual prediction that avoids explicit structural causal model (SCM) estimation. Under a monotonicity assumption on $Y = f(X, Z, r(E))$, it argues that the factual outcome identifies a conditional quantile level that is preserved under intervention, so the counterfactual can be recovered from the corresponding conditional quantile at $X = x'$. The method is implemented as a neural bi-level model that predicts both the instance-specific quantile level and the associated conditional quantile value. The idea is interesting, the formulation is clean, and the empirical results are strong on the reported benchmarks. My main reservation is that the paper's theoretical and empirical claims are broader than what is actually established for the practical method.

**Additional Comments:**

I think the paper has a real idea and promising results. My concern is mainly overclaiming as of now: the current draft presents partial and population-level results as if they were guarantees for the practical method. Tightening the assumptions and claims would substantially improve the paper.

**Audience:**

Yes

**Audience Explanation:**

Yes. The quantile-preservation view of counterfactual prediction is novel and relevant to researchers in causal ML, identifiable generative models, and distributional prediction, even if the current draft seems to overstate what is proven, again until further clarifications from the authors.

**Broader Impact Concerns:**

I do not have a paper-specific broader-impact concern that would require a dedicated statement. More generally, the authors should avoid implying reliability outside the assumptions used for identifiability, especially in settings with latent confounding, non-monotonic mechanisms, or high-stakes decisions.

**Claims And Evidence:**

No

**Claims Explanation:**

Until further clarifications from the authors, in my view the main issue is the gap between the theory and the implemented method. Theorem 1 is a population-level result for the idealized bi-level problem, and its proof effectively assumes that the lower-level function class recovers the true conditional quantile function. That assumption should be stated explicitly. More importantly, this does not justify the actual neural procedure in Equations (9)-(11), which uses parameterized models, alternating optimization, and approximate hypergradients.

The interpretation of Theorem 2 is also too strong, it seems to me. The result bounds an excess-risk term for the learned quantile regressor at the estimated quantile level, but it is not a finite-sample guarantee for the full counterfactual predictor: it does not account for error in estimating $\hat{\tau}$, model misspecification, or optimization error. The paper repeatedly presents it as a reliability guarantee for counterfactual prediction. There also appears to be an inconsistency between Equation (13) and Equation (30): the former has $2b / \sqrt{N}$, while the proof ends with $4b / \sqrt{N}$?

I was also not totally convinced by the "implicit monotonicity" claim in Section 4.2. Pinball loss alone does not guarantee non-crossing quantiles in a flexible neural model. Empirically, the results are good, but they are mostly on synthetic or semi-synthetic settings aligned with the assumptions, so they provide only partial support for the broader claims. The image experiments are interesting, but the paper should be more precise about what quantile object is being estimated in the high-dimensional setting.

**Requested Changes:**

Critical for my recommendation, even though I look forward to authors' clarifications:

1. Clarify the assumptions behind Lemma 1 and Theorem 1, especially what is required of the function class $F$ and how the population-level theory relates to the neural algorithm in Equations (9)-(11).
2. Recheck Theorem 2 and Appendix E. Equation (13) appears inconsistent with Equation (30), and the interpretation of the bound should be narrowed unless the authors can justify a guarantee for the full learned counterfactual predictor.
3. Weaken or justify the "implicit monotonicity" claim in Section 4.2. As written, it seems to overstate what the training objective guarantees.
4. Clarify the formal meaning of the quantile object in the high-dimensional image setting.
5. Tone down claims about uniqueness, reliability, and broad applicability in the abstract, theory section, and conclusion unless stronger support is added.

Changes that would strengthen the paper:

1. Add a simpler baseline closer to the proposed decomposition, such as a conditional CDF plus quantile estimator or a discretized quantile-search baseline.
2. Clarify the model-selection and hyperparameter-selection protocol for the proposed method and baselines.
3. Improve presentation of the theory and tables; some proof steps are compressed and some uncertainty values in Table 1 appear truncated.

---

> ### Author Response · Authors · 2026-05-05
> **Response to Reviewer k5Rv**
>
> Thank you for your appreciation of our method as being interesting, clean, and delivering strong performance. We appreciate your positive feedback. We have carefully followed your constructive comments to further clarify our contributions and improve the clarity and precision of the manuscript. Below, we provide detailed responses to your questions. Please feel free to let us know if you have any further questions or suggestions.
>
> ___
>
> ### the lower-level function class and Idealized analysis and neural implementation
>
> We thank the reviewer for this important comment regarding the gap between the theoretical result and the implemented method. We agree with you that theorem1 is under population-level. Therefore,
> 1. we have added "Population-level analysis" in Section 4.3 before the theorem1.
>
> 2. The reviewer is absolutely correct that the proof of Theorem 1 implicitly assumes that the lower-level function class is expressive enough to recover the true conditional quantile function. We apologize for leaving this implicit. To correct this, we have added the assumption the theorem.1
> >and assuming that the true conditional quantile function is contained within the lower-level hypothesis class $\mathcal{F}$
>
>
> 3. We appreciate the reviewer highlighting the gap between the idealized bi-level problem and the empirical neural procedure (Eqs. 9-11). You are entirely correct that Theorem 1 does not guarantee the convergence of parameterized neural models using alternating optimization and approximate hypergradients.
>
> Our intention with Theorem 1 was to justify the formulation of the objective itself—proving that the bi-level formulation is mathematically sound and that its global minimizer uniquely and correctly identifies the true counterfactual. We did not intend to claim that it proves the convergence dynamics of the neural optimizer. Proving global convergence for non-convex deep neural networks using approximate hypergradients remains a notoriously difficult open problem in deep learning theory.
>
> To ensure we do not overstate our theoretical claims, we have added a dedicated paragraph explicitly discussing this gap, positioning Theorem 1 as a representational guarantee rather than an optimization guarantee, and relying on our empirical results to demonstrate the efficacy of the approximate hypergradients
> > Remark on Theory vs. Implementation: We note that Theorem 1 characterizes the global minima of the idealized, population-level bi-level objective. It guarantees that our formulation is mathematically sound: if the global minimum is reached, it yields the true counterfactual. However, it does not provide convergence guarantees for the empirical procedure detailed in Eqs. 9–11. In practice, we rely on parameterized neural networks, alternating optimization, and approximate hypergradients. While bridging this optimization gap for non-convex architectures remains an open challenge, our empirical results in Table 1 demonstrate that this approximate procedure performs well in our evaluated settings, indicating that the theoretical formulation translates effectively to practical implementation.

---

> > ### Author Response · Authors · 2026-05-05
> > **Response to Reviewer k5Rv - Continued**
> >
> > ---
> >
> > ### The interpretation of Theorem 2
> >
> >
> >
> >
> > We sincerely thank the reviewer for their rigorous evaluation of Theorem 2 and for catching the inconsistency in our equations. We completely agree with your assessment and have made revisions to ensure our claims are more precise:
> >
> > 1. We have removed wording that framed Theorem 2 as a “reliability guarantee for counterfactual prediction” and instead describe it more precisely as a “generalization bound for the conditional quantile regression component” in Theorem 2.
> >
> >
> >
> > 2. We have added a dedicated paragraph to the end of Section 4 to explicitly acknowledge the gap between our theoretical bounds and the end-to-end empirical procedure. The newly added text reads as follows:
> > > "However, we note that Theorem 2 provides a guarantee for the conditional quantile regression component at the estimated quantile level, rather than a finite-sample guarantee for the full, end-to-end counterfactual predictor. Establishing a strict global bound is highly non-trivial, as it would require systematically accounting for the estimation error of the upper-level quantile level $\tau$, potential model misspecification, and the optimization error induced by the alternating training procedure. While such a comprehensive theoretical analysis remains an important direction for future work, our framework proves highly effective in practice. Complementing the theoretical insights of Theorem 2, we empirically demonstrate in Section 5.3 that our method achieves significantly better performance than strong baselines given the same sample size. In summary, our framework provides solid theoretical grounding for the conditional quantile regression component, which successfully translates into robust empirical performance for counterfactual prediction."
> >
> > 3. Thanks for pointing out the typo in Theorem 2. It should be $4b/\sqrt{N}$, we have corrected it in theorem 2.
> >
> >
> > ___
> >
> > ### implicit monotonicity
> >
> > We thank the reviewer for this rigorous mathematical point. We completely agree: optimizing a flexible neural network with the empirical pinball loss does not strictly guarantee non-crossing quantiles. Our use of the phrase 'implicit monotonicity' in Section 4.2 was overly broad and pertained to our empirical observations rather than a theoretical guarantee. To ensure precision, we have revised this paragraph to clarify that we opted against strict constraints to avoid optimization complexity and preserve model expressivity, supported by our empirical observation that severe quantile crossing does not occur in practice:
> > >Practical considerations regarding monotonicity. Theoretically, our bi-level formulation relies solely on the pinball loss without incorporating external regularization. Consequently, it does not strictly prevent issues such as quantile crossing in highly flexible neural networks. We opt not to enforce strict monotonicity primarily to avoid the optimization complexity and reduced expressivity associated with rigid architectural constraints, such as those used in BGM (Nasr-Esfahany et al., 2023). Empirically, however, we observe that severe quantile crossing does not occur in our evaluated settings (see, for example, Fig. 7 and Fig. 4).Nevertheless, if a specific application requires strict adherence to the monotonicity assumption, one could readily extend our framework by incorporating explicit non-crossing penalties (e.g., Tagasovska & Lopez-Paz,2019) or by utilizing inherently monotonic network architectures.
> >
> >
> >
> > ### quantile objective in high-dimensional
> >
> > We thank the reviewer for highlighting the need for clarity regarding our high-dimensional experiments. To clarify, we use the same fundamental bi-level objective for the image datasets as we do for the tabular data, but we evaluate the quantiles on a marginal, pixel-wise basis.Specifically, we adapt our two-level architecture by replacing the MLPs with CNNs. The upper-level network still outputs a single scalar $\tau$ (representing the individual's underlying rank/noise), while the lower-level network outputs the high-dimensional image. Because standard quantiles are not naturally defined over joint high-dimensional spaces, the quantile object being estimated is the pixel-wise marginal conditional quantile. To ensure this is perfectly clear to the reader, we have updated Section 5.4 to explicitly detail this design. The added text reads as follows:
> > > The quantile regression objective (pinball loss) is applied element-wise across all pixels, and then averaged to form the final lower-level loss.

---

> > > ### Author Response · Authors · 2026-05-05
> > > **Response to Reviewer k5Rv - Continued**
> > >
> > > ### conclusion claims
> > > We thank the reviewer for this suggestion. To ensure precision and avoid overclaiming our theoretical contribution, we have updated the conclusion to provide a direct characterization of our technical approach. Specifically, we removed the broader theoretical claims ("In this paper, we approach this problem... under mild conditions.") and replaced them with the following concise statement:
> > > > In this paper, we develop a neural bi-level optimization framework that jointly estimates the quantile function and the corresponding quantile level.
> > >
> > > ___
> > >
> > > ### simpler baseline
> > > We sincerely thank the reviewer for this constructive suggestion. To address this, we have added a "Discretized Quantile Search" baseline, which serves as a direct ablation of our own method. Specifically, we retain our proposed neural quantile architecture trained with the pinball loss over randomly sampled quantiles. During inference, we perform a decoupled grid search over $\tau \in [0, 1]$ (with a 0.01 interval) to select the value that best reconstructs the factual outcome.
> > >
> > > We have updated Table 1 to include these results (also presented below for completeness). We find that while this simplified version of our method (our neural pinball implementation combined with grid search) already outperforms several standard baselines, it remains inferior to our full proposed approach. This empirical performance gap directly highlights the necessity and advantage of our joint bi-level formulation over a decoupled, discrete search.
> > >
> > >
> > >
> > > | Method | IHDP Train | IHDP Test | Cont-Dose Train | Cont-Dose Test | Dis-Dose Train | Dis-Dose Test |
> > > |--------|------------|-----------|------------------|----------------|----------------|----------------|
> > > | **Discretized Quantile Search** | **1.37 ± .2** | **1.35 ± .2** | **0.08 ± .0** | **0.08 ± .0** | **0.27 ± .0** | **0.28 ± .0** |
> > > | **Ours** | **1.29 ± .3** | **1.23 ± .2** | **0.06 ± .0** | **0.06 ± .0** | **0.20 ± .0** | **0.20 ± .0** |
> > >
> > > We have added the following discussion to Section 5.1 to discuss this:
> > >
> > > >We further confirm this necessity through an ablation study (Table 1), where we replace our bi-level formulation with a discrete grid search ($\tau \in [0, 1]$ at $0.01$ intervals) over our trained neural quantile regressor. Although this discretized baseline is highly competitive, our full bi-level approach significantly outperforms it. This highlights that optimizing the quantile estimation and counterfactual prediction jointly, rather than sequentially, is crucial for maximizing performance.
> > >
> > >
> > > ___
> > >
> > > ### Clarify the model-selection and hyperparameter-selection protocol for the proposed method and baselines.
> > >
> > > We thank the reviewer for the opportunity to clarify our experimental protocol. For all baseline methods, we utilize the optimal hyperparameters recommended by their respective authors. For our proposed method, we first perform hyperparameter tuning using simulated datasets—specifically, the five commonly used SCMs shown in Fig. 4—because these provide ground-truth counterfactuals that allow for rigorous evaluation. We then reuse these tuned configurations for the benchmark and high-dimensional datasets. During training on these datasets, final model selection is determined strictly based on the reconstruction performance of the factual samples. Finally, regarding the network architectures, we intentionally avoided heavy engineering. We adopted standard, off-the-shelf MLP and CNN structures to highlight that our performance gains stem fundamentally from our proposed bi-level formulation rather than specialized network designs.
> > >
> > > We thank the reviewer again for your positive feedback and constructive suggestions regarding the rigor of our claims and theoretical results. We believe these revisions have significantly improved the clarity and robustness of our manuscript.

---

### Review · Reviewer_BES8 · 2026-04-20

**Summary Of Contributions:**

The paper proposes a neural bi-level quantile-regression framework for individual counterfactual prediction. Its central claim is that, under a causal mechanism for the outcome $Y$ of the form $Y=f(X,Z,r(E))$, with $f$ smooth and strictly monotone in the transformed noise $r(E)$, a factual observation and its counterfactual share the same conditional quantile level. The method then learns that latent quantile level with one network and the conditional quantile function with another, and reports strong RMSE improvements over several deep-SCM baselines on synthetic, semi-synthetic, and constructed image datasets.

**Audience:**

No

**Audience Explanation:**

In its current form, I do not think the paper would be sufficiently interesting to TMLR's audience because its main conceptual idea is not positioned against the closest prior literature. The manuscript presents the quantile-preservation view as a novel connection between counterfactual prediction and quantile regression, but it does not engage with the optimal-transport literature where closely related quantile-based and transport-based counterfactual constructions are already developed. As a result, the true incremental contribution appears limited.

The paper could in principle focus on presenting new neural bilevel estimator for counterfactual outcomes under monotone latent ordering. That is, building a new estimation framework that sits on top of the current 1D/quantile/rank-preserving counterfactual maps in the literature. Yet, that would require the paper to be fully written from scratch.

**Broader Impact Concerns:**

I have no concerns.

**Claims And Evidence:**

No

**Claims Explanation:**

The paper provides empirical evidence within its own assumed regime, especially on synthetic and semi-synthetic settings where the monotonic latent-ordering assumption is built into the data-generating process. The quantile-preservation mechanism is clearly explained, and the experiments show that the proposed estimator can work well when those assumptions hold.

However, the evidence is not fully convincing for the paper's stronger claims about novelty and identifiability. In particular, the related-work section omits closely related optimal-transport and rank/quantile-preserving counterfactual literature, which has grown substantially in the last years. It is my understanding that the core 1D construction (via quantile regression) is exactly of the form of many other papers that were not referenced, for instance:

- Plečko, D., and Meinshausen, N. (2020). *Fair Data Adaptation with Quantile Preservation*. _Journal of Machine Learning Research_, 21(243): 1–44.
- Charpentier, A., Flachaire, E., and Gallic, E. (2023). *Optimal Transport for Counterfactual Estimation: A Method for Causal Inference*. In: Ngoc Thach, N., Kreinovich, V., Ha, D.T., Trung, N.D. (eds) _Optimal Transport Statistics for Economics and Related Topics. Studies in Systems, Decision and Control_, vol 483. Springer
- De Lara, L., González-Sanz, A., Asher, N., Risser, L., and Loubes, J.-M. (2024). *Transport-based Counterfactual Models*. _Journal of Machine Learning Research_, 25(136): 1–59.
- Fernandes Machado, A., Charpentier, A., and Gallic, E. (2025). *Sequential Conditional Transport on Probabilistic Graphs for Interpretable Counterfactual Fairness*. _Proceedings of the AAAI Conference on Artificial Intelligence_, 39(18).

**Requested Changes:**

I would recommend the authors to:
- Substantially revise the related-work section. The paper should discuss the closest optimal-transport and quantile-preserving counterfactual literature, and clearly explain what is new here relative to quantile-preserving transport and related counterfactual constructions.
- Reframe the contribution more narrowly and accurately. A more defensible claim would be that the paper introduces a new neural bilevel estimator for counterfactual prediction under monotone latent ordering, rather than a broadly new theoretical paradigm for counterfactual prediction, which has already been presented in the literature.

---

> ### Author Response · Authors · 2026-05-05
> **Response to Reviewer BES8**
>
> We thank the reviewer for your constructive and insightful feedback. We particularly appreciate the suggestion to reframe our contribution more narrowly and accurately as a new **neural bi-level estimator**. We agree that this positioning better reflects the incremental contribution of our work relative to the established literature on quantile preservation and optimal transport.
>
> Below, we first provide a discussion on the methodological connectionss between our approach and the referenced works, followed by a summary of the specific revisions made to the manuscript.
>
>
> ### Relationship to Prior Work
>
>
> We thank the reviewer for pointing out the important works [1, 2, 3, 4]. We have carefully read them and agree they are essential for contextualizing our work.
>
> *  **Shared Principles** These works share the high-level principle that preserving quantiles can enable counterfactual predictions, either explicitly or implicitly, particularly in the one-dimensional setting. Their success across different applications, such as fairness [1,3,4], further supports the validity of our approach, which is largely grounded in this principle. We would also like to clarify that we did not intend to present this principle as a fully novel contribution. After Lemma 1, we discuss several closely related works, including Nasr-Esfahany et al. (ICML 2023) and RankPrev (NeurIPS 2025), that reflect similar ideas. At the same time, we acknowledge that some of our wording may have inadvertently overstated the contribution. We thank the reviewer for pointing this out, as well as for highlighting additional relevant literature, particularly from the optimal transport perspective, which helps further refine and strengthen the positioning of our paper.
>
>
> * **Our main contribution** As highlighted by the reviewer, our primary contribution is a neural bi-level estimator for counterfactual outcomes under monotone latent ordering. While the underlying quantile-preservation principle is established, we are, to the best of our knowledge, among the first to formally cast counterfactual prediction as a bi-level quantile regression problem. In addition, we provide theoretical analysis of this bi-level framework, which is not addressed in the referenced works [1,2,3,4].

---

> > ### Author Response · Authors · 2026-05-05
> > **Response to Reviewer BES8 - Continued**
> >
> > ### Summary of Revisions (Highlighted in Red in the Manuscript)
> >
> > In light of your valuable suggestions, we have reframed the paper to focus on the neural estimation framework rather than a new theoretical paradigm.
> >
> > 1. In the Abstract and Section 3, we have replaced claims regarding "novel connections" with a more precise methodological focus:
> >
> > > "We show that counterfactual prediction can be reframed as an extended bi-level quantile regression problem."
> >
> > 2. We have updated the Introduction to better contextualize our work within the existing literature and further clarify our primary contributions:
> > > While this theoretical insight is intuitive, effectively operationalizing it efficiently remains an open challenge. Although similar rank-preserving principles have been shared by previous works (Chernozhukov & Hansen, 2005; Plečko & Meinshausen, 2020; Charpentier et al., 2023; De Lara et al., 2021; Machado et al., 2025; Wu et al., 2025), these methods face significant practical limitations. Specifically, they often assume access to the underlying density functions of the data-generating mechanism (Plečko & Meinshausen, 2020), rely on inaccurate high-dimensional density estimation (Wu et al., 2025) (see Table.1), or utilize optimal transport formulations where “implementation is challenging (except in the Gaussian case), and usually hard to interpret” (Machado et al., 2025).
> >
> > 3. In contribution, we replace "This framework is grounded in a {formal connection between counterfactuals and conditional quantiles}, allowing us to identify counterfactual outcomes by preserving the quantile level under mild assumptions" with
> > > By operationalizing the established connection between counterfactuals and conditional quantiles, our framework estimates counterfactual outcomes via quantile preservation under standard assumptions.
> >
> >
> > 5.  We added following in Related work and here we quote:
> > > Several works share the core principle of preserving quantiles or utilizing transport mappings for counterfactual prediction. Chernozhukov & Hansen (2005); Chernozhukov et al. (2017) establish rank invariance to identify quantile treatment effects using instrumental variables and explicit structural equations. However, their methodology relies on an inverse inference procedure that typically requires a grid search over the parameter space, which is computationally prohibitive in high-dimensional settings. In contrast, we propose a direct neural bi-level optimization framework. Our approach learns quantile levels directly from observational data without specifying a structural model, yielding precise, point-valued individual counterfactual predictions via scalar quantile preservation. Plečko & Meinshausen (2020) propose quantile preservation but assume access to the underlying data-generating densities. Alternatively, Charpentier et al. (2023), De Lara et al.(2021), and Machado et al. (2025) rely on optimal transport theory—using it to justify quantile matching, formulate probabilistic distributional couplings, or construct sequential causal mappings, respectively. Instead of relying on assumed densities or explicit optimal transport constructions, we propose a direct neural bi-level estimation framework. Our approach learns quantile levels directly from data without specifying the model, yielding precise, point-valued counterfactual predictions via scalar quantile preservation."
> >
> > 3. In the discussion following Lemma 1 in Section 3, we added:
> >
> > > Our result provides a direct characterization of counterfactual outcomes in terms of conditional quantiles. Some optimal transport approaches also implicitly preserve quantile or rank information (Charpentier et al., 2023; De Lara et al., 2024), but they typically rely on distributional modeling or transport constructions that can be challenging to implement and interpret in complex settings (Machado et al.,2025). Other methods also characterize counterfactual outcomes, but often require access to the underlying data-generating densities (Plečko & Meinshausen, 2020) or rely on instrumental variables (Chernozhukov & Hansen, 2005; Chernozhukov et al., 2017).
> >
> >
> >
> >
> > We sincerely thank you again for pointing us to these important references and for your thoughtful guidance on positioning our work. Your suggestions have greatly helped improve the clarity, accuracy, and overall completeness of our paper.

---

> > > ### Author Response · Authors · 2026-05-05
> > > **Response to Reviewer BES8 - Continued**
> > >
> > > [1] Plečko, D., and Meinshausen, N. (2020). Fair Data Adaptation with Quantile Preservation. Journal of Machine Learning Research, 21(243): 1–44.
> > >
> > > [2] Charpentier, A., Flachaire, E., and Gallic, E. (2023). Optimal Transport for Counterfactual Estimation: A Method for Causal Inference. In: Ngoc Thach, N., Kreinovich, V., Ha, D.T., Trung, N.D. (eds) Optimal Transport Statistics for Economics and Related Topics. Studies in Systems, Decision and Control, vol 483. Springer
> > >
> > > [3] De Lara, L., González-Sanz, A., Asher, N., Risser, L., and Loubes, J.-M. (2024). Transport-based Counterfactual Models. Journal of Machine Learning Research, 25(136): 1–59.
> > >
> > > [4] Fernandes Machado, A., Charpentier, A., and Gallic, E. (2025). Sequential Conditional Transport on Probabilistic Graphs for Interpretable Counterfactual Fairness. Proceedings of the AAAI Conference on Artificial Intelligence, 39(18).

---

### Decision · Action_Editor_zVNG · 2026-06-01

**Recommendation:** Accept with minor revision

**Additional Comments:**

Between them, reviewers pointed out several contributions presented in this submission that will be of interest to TMLR's audience. The primary concern in the evaluation was the relationship between the submission and previous work that connects quantile regression to counterfactual prediction. While the authors substantially revised parts of the paper to better reflect this relationship, the framing of the abstract and introduction is still based on the relation between traditional approaches (estimating the full SCM) and the conceptual innovation of considering quantile regressions instead (e.g., Section 1, paragraph 3), rather than the remaining shortcomings of other methods based on quantiles. As a reviewer pointed out, the way the statement is made is not technically incorrect, but it may mislead the reader as to the novelty of the submission. For example, in the added paragraph in the revision, "quantile" is not even mentioned, despite several of the cited works using it in their titles.

I request that the introduction be revised to make it clearer what part of the conceptual innovation was already made in previous work, and what gaps remain that are addressed by the current submission.

**Audience:**

Yes

**Audience Explanation:**

Reviewers mostly appreciated the contributions of this work, and its topic lies firmly within the areas of interest for TMLR. Concerns were raised that the audience's interest in the paper would be smaller in light of related work that was not originally discussed in the submission. This has been improved in revisions by the authors, and I have suggested that this will be clarified further in the final version.

**Claims And Evidence:**

Yes

**Claims Explanation:**

All reviewers agreed that the submission is supported by accurate, convincing and clear evidence.